# Optimal Query Complexity of Secure Stochastic Convex Optimization

**Wei Tang[†], Chien-Ju Ho[†], and Yang Liu[*]**
[†]Washington University in St. Louis, [*]UC Santa Cruz
{w.tang, chienju.ho}@wustl.edu, yangliu@ucsc.edu

## Abstract

We study the *secure* stochastic convex optimization problem. A learner aims to learn the optimal point of a convex function through sequentially querying a (stochastic) gradient oracle. In the meantime, there exists an adversary who aims to free-ride and infer the learning outcome of the learner from observing the learner's queries. The adversary observes only the points of the queries but not the feedback from the oracle. The goal of the learner is to optimize the accuracy, i.e., obtaining an accurate estimate of the optimal point, while securing her privacy, i.e., making it difficult for the adversary to infer the optimal point. We formally quantify this tradeoff between learner's accuracy and privacy and characterize the lower and upper bounds on the learner's query complexity as a function of desired levels of accuracy and privacy. For the analysis of lower bounds, we provide a general template based on information theoretical analysis and then tailor the template to several families of problems, including stochastic convex optimization and (noisy) binary search. We also present a generic secure learning protocol that achieves the matching upper bound up to logarithmic factors.

## 1  Introduction

Optimization, that seeks to find the optimal point of a function, is an important tool in various domains, including decision making and machine learning. Modern optimization techniques, such as gradient descent, often run in an iterative manner: the learner adaptively queries a (noisy) oracle, obtains the information about the function (e.g., gradient) at the query point, and updates the estimate of the optimal point. While such iterative techniques have been well studied and shown to be efficient, the iterative nature introduces potential risks of information leak. A spying adversary, who can observe the series of query points the learner sends to the oracle but not the oracle responses, may free-ride and infer the optimal point from the queries alone.

For example, consider a company aiming to find the optimal price for a new product. The company might hire market research firm that performs dynamic pricing on a test population. Assume the market research firm is adopting an optimization algorithm that increases the price if the sale happens and decreases the price otherwise. An adversary (e.g., a competing company), who knows the algorithm and can observe the price changes (e.g., by entering the test population), may infer and estimate the optimal price before the product launch even without knowing whether the transaction happens or not during market research. As another example, in federated learning, the learner might aim to optimize the parameters of their learning models using gradient decent. Since data might be distributed, the learner needs to sequentially broadcast their models to data-holding users in order to obtain the gradient information. An adversary can pretend to be data-holding user to receive the

sequence of broadcasted models. He might then estimate the final model even without obtaining the gradient information.

In this work, we study the *secure* stochastic convex optimization problem, in which the learner aims to optimize the *accuracy*, i.e., obtain an accurate estimate to the optimal point, while securing her *privacy*, i.e., preventing an adversary from inferring what she learned[1]. We formalize the notions of accuracy and privacy using PAC (Probably Approximate Correct) style notions. The algorithm is $(\epsilon, \delta)$-accurate if the learner's estimate is within $\epsilon$ distance to the optima with probability at least $1 - \delta$. The algorithm is $(\epsilon^{\mathrm{adv}}, \delta^{\mathrm{adv}})$-private[2] if for any adversary that can infer from only the query points, the probability for his estimate to be within $\epsilon^{\mathrm{adv}}$ distance to the optima is at most $\delta^{\mathrm{adv}}$. Our goal is to characterize the trade-offs between learner's accuracy and privacy using query complexity, i.e., the minimum number of queries needed to achieve a given level of accuracy and privacy.

Our main results include the characterization of the lower and upper bounds of the query complexity for the secure stochastic convex optimization problem. In particular, we study the general $\kappa$-uniformly convex functions. We show that, with logarithmic factors compressed in the bounds, when the error measure is function error (i.e., the error is the difference of the objective function values between the estimate and the optima), we obtain matching upper and lower bounds in the order of $\Theta\left(1/(\delta^{\mathrm{adv}}\epsilon^{(2\kappa-2)/\kappa})\right)$. When the error measure is point error (i.e., the error is the difference between the estimate and optima in the input domain), we obtain matching upper and lower bounds in the order of $\Theta\left(1/(\delta^{\mathrm{adv}}\epsilon^{2\kappa-2})\right)$. Our results recover the classic complexity bounds in convex optimization (strongly convex for $\kappa = 2$ and convex for $\kappa \to \infty$) when there is no requirement to secure the learner's privacy. Our bounds suffer an additional factor of $\Theta(1/\delta^{\mathrm{adv}})$ compared to classic non-secure bounds[3], which can be viewed as a complexity price that the learner has to pay to secure her privacy.

To highlight our technical contributions, for the lower-bound analysis, we develop a general template based on an information-theoretical analysis for convex programming [11]. In addition to deriving the lower bound, we demonstrate that the same template can be applied to obtain the same lower bound of private binary search [20], in which the authors focus on a (Bayesian) binary search problem and assume the learner has a uniform prior on where the target is and has access to a *noiseless* oracle. In addition to obtaining the same lower bound using different techniques, we show that the template offers the lower bound for private *noisy* binary search, which has been also discussed in a recent work [19]. As for the upper bound, we propose a secure learning protocol that is immune to any adversary. The protocol may incorporate an arbitrary non-secure but efficient learning algorithm as a subroutine, and a matching upper bound up to logarithmic factors is proved.

**Related work.**     This paper is closely related to the recent works in private sequential learning [20, 17, 19], which study private Bayesian binary search: A learner aims to estimate an unknown target value through sequentially querying an oracle which returns exact binary responses, while protecting her estimations from an adversary. The authors assume that the learner has a uniform prior for the unknown target value. We generalize their setting of binary search to stochastic convex optimization and adopts different analysis which builds on minimax bounds instead of assuming uniform prior.

Another close line of research is differentially private online learning [2, 3, 5, 6, 8, 15, 16]. Our work departs significantly from these works. In differential privacy, the goal is to ensure the change for any individual participant does not change the outcome substantially, and therefore the privacy of individuals is protected. The goal of our work is to secure the learner's privacy in the sense that the adversary cannot infer what the learner is learning from observing the actions of the learner. We name our work *secure* optimization (where the learner's objective is secured from the adversary) to

emphasize this difference. Our technique is built on the minimax analysis for (stochastic) convex optimization problem [1, 4, 7, 9–11, 13, 14]. Our results complement this line of work through incorporating the privacy requirement.

## 2  Problem Formulation

Consider a *learner* $\mathcal{A}$ who aims to maximize the *accuracy* of learning the optimal point of an unknown convex function $f$ through sequentially querying an oracle $\phi$ about the function information. In the meantime, the learner wants to secure her *privacy*, i.e., preventing a spying *adversary* from free-riding and inferring the learning outcome through observing where the learner queries. A problem class of convex optimization problem is defined by a triple $\mathcal{P} = (\mathcal{X}, \mathcal{F}, \phi)$, where $\mathcal{X} \subset \mathbb{R}^d$ is a compact and convex problem domain, $\mathcal{F}$ is a class of convex functions, and for any function $f \in \mathcal{F}$, $\phi : \mathcal{X} \times f \to \mathcal{Y}$ is an oracle function that answers any query $x \in \mathcal{X}$ by returning an element $\phi(x, f)$ in an information set $\mathcal{Y}$.

At the beginning of the learning process, an unknown convex objective function $f$ is drawn from $\mathcal{F}$. Let $x_f^*$ be the minimizer of $f$, i.e., $x_f^* = \arg\min_{x \in \mathcal{X}} f(x)$, and $f^* = f(x_f^*)$ be the optimal function value. At each time $t = 1$ to $T$, the learner submits a query $X_t$ to the oracle and obtains a response $Y_t = \phi(X_t, f)$. Let $X^T = \{X_1, \ldots, X_T\}$ denote the set of queries till time $T$. Similarly $Y^T = \{Y_1, \ldots, Y_T\}$ denotes the set of corresponding responses. The learner can observe all queries and responses, i.e., $X^T$ and $Y^T$, while the adversary can only observe the queries $X^T$. At the end of the learning, the learner outputs an estimate $\widehat{X}$ for $x_f^*$, based on $X^T$ and $Y^T$, while the adversary outputs another estimate $\widehat{X}^{\mathrm{adv}}$ based only on the query points $X^T$ but not the responses.

**Objective.**   The learner aims to design an algorithm $\mathcal{A}$, which sequentially decides $X_t$ and formulates a candidate optimizer $\widehat{X}$ (optimizer here is equivalent to the estimate), with the goal of minimizing the number of queries while ensuring *accuracy*, i.e., $\widehat{X}$ is a good estimate to the optimal point $x_f^*$, and securing *privacy*, i.e., $\widehat{X}^{\mathrm{adv}}$ is sufficiently far away from $x_f^*$ for any adversary.

We use $\mathrm{err}(\widehat{X}, f)$ to measure how close an estimate $\widehat{X}$ is to the optimal point of function $f$. Two generic error measures are: function error $\mathrm{err}(\widehat{X}, f) = |f(\widehat{X}) - f^*|$ and point error $\mathrm{err}(\widehat{X}, f) = \|\widehat{X} - x_f^*\|$, where $\| \cdot \|$ denotes the Euclidean norm. With the error measure in place, we formally define the notions of learner's accuracy and privacy requirements:

**Definition 1** (($\epsilon, \delta$)-accurate). Fix $\epsilon, \delta \in (0, 1)$. Given a problem class $\mathcal{P} = (\mathcal{X}, \mathcal{F}, \phi)$, a learner's algorithm $\mathcal{A}$ is ($\epsilon, \delta$)-accurate if for any $f \in \mathcal{F}$,

$$\mathbb{P}(\mathrm{err}(\widehat{X}, f) \geq \epsilon) \leq \delta, \tag{1}$$

where the probability is measured with respect to the randomness in the oracle's responses and the possible randomness in the algorithm.

**Definition 2** (($\epsilon^{\mathrm{adv}}, \delta^{\mathrm{adv}}$)-private). Fix $\epsilon^{\mathrm{adv}}, \delta^{\mathrm{adv}} \in (0, 1)$. A learner's algorithm is ($\epsilon^{\mathrm{adv}}, \delta^{\mathrm{adv}}$)-private if, for any adversary estimator $\widehat{X}^{\mathrm{adv}}$ and for any $f \in \mathcal{F}$,

$$\mathbb{P}(\mathrm{err}(\widehat{X}^{\mathrm{adv}}, f) \leq \epsilon^{\mathrm{adv}}) \leq \delta^{\mathrm{adv}}, \tag{2}$$

where the probability is measured with respect to the randomness in the oracle's responses, the algorithm, and the adversary estimator.

**Remark 1.** We choose to use the term "private" here in the definition in the sense that the algorithm aims to secure the privacy of the learner.

Intuitively, an algorithm is ($\epsilon, \delta$)-accurate if the estimate $\widehat{X}$ is within $\epsilon$ distance to the optima with probability at least $1 - \delta$, and an algorithm is ($\epsilon^{\mathrm{adv}}, \delta^{\mathrm{adv}}$)-private if for any adversary, with probability at most $\delta^{\mathrm{adv}}$, the estimate $\widehat{X}^{\mathrm{adv}}$ is within $\epsilon^{\mathrm{adv}}$ to the optima.

The goal of the learner is to minimize the number of queries while satisfying the requirements of achieving a given level of accuracy and securing her privacy. To characterize this goal, we define *secure query complexity* as follows:

**Definition 3** (Secure Query Complexity). Given a problem class $\mathcal{P} = (\mathcal{X}, \mathcal{F}, \phi)$, the secure query complexity $T_{\mathcal{P}}(\epsilon, \delta, \epsilon^{\mathrm{adv}}, \delta^{\mathrm{adv}})$ is defined as the least number of queries needed for a learner's algorithm to be simultaneously $(\epsilon, \delta)$-accurate and $(\epsilon^{\mathrm{adv}}, \delta^{\mathrm{adv}})$-private for any function $f \in \mathcal{F}$.

When it is clear from the context, we drop the input parameters and simply write $T_{\mathcal{P}}$.

## 2.1 Problem Classes

We illustrate the problem classes $\mathcal{P} = (\mathcal{X}, \mathcal{F}, \phi)$ that we explore in this work.

**Types of oracle.** We focus on settings in which the oracle returns the first-order information (as is common in gradient-based optimization algorithms). In particular, let $g(x)$ be an arbitrary subgradient in $\partial f(x)$. If the oracle only returns the sign of $g(x)$, we denote such oracle by $\phi^{\mathsf{sign}}$. A noisy sign oracle with correct probability being $p \in (0.5, 1)$ will be denoted by $\phi^{\mathsf{sign},p}$. We also consider the standard noisy first-order oracle that returns noisy 0-th and 1-st order information, where the information consists of the pair $(f(x) + Z_1, g(x) + Z_2)$, with the noise $Z_1$ added to the function value being drawn from $\mathcal{N}(0, \sigma^2)$ (zero-mean Gaussian distribution) and the noise $Z_2$ to the first-order information being drawn from $\mathcal{N}(0, \sigma^2 \mathcal{I}_d)$. We use $\phi^{(1)}$ to denote such noisy first-order oracle and refer it as the Gaussian oracle.

**(Noisy) binary search.** One of the simplest setups of our framework is the one-dimensional binary search, in which $\mathcal{X} = [0, 1]$, $\mathcal{F}^{\mathrm{Abs}} = \left\{ f(x) \triangleq |x - x^*| \right\}$, and the oracle is $\phi^{\mathsf{sign}}$ (i.e., whether the query $x$ is larger than the optimal $x^*$). The above setting can extend to a noisy binary search with oracle $\phi^{\mathsf{sign},p}$.

**Convex optimization.** We also explore the general convex optimization problem with first-order oracle $\phi^{(1)}$. We consider the general class of $\kappa$-uniformly convex function. Given $\kappa \geq 2$, let $\mathcal{F}^{\kappa}$ be the set of all convex functions that satisfy: $f(x) - f(x_f^*) \geq \frac{\lambda}{2}\|x - x_f^*\|^{\kappa}, \forall x \in \mathcal{X}$, for some $\lambda > 0$. $\kappa$-uniformly convex function is a general representation of convex functions: when $\kappa = 2$, it recovers strong convexity, and when $\kappa \to \infty$, it recovers (non-strong) convexity. We shall always assume the functions in $\mathcal{F}^{\kappa}$ are $L$-Lipschitz, i.e., for all $f \in \mathcal{F}^{\kappa}$ and all $x, y \in \mathcal{X}$, $\|f(x) - f(y)\| \leq L\|x - y\|$.

# 3 Lower Bounds on Secure Query Complexity

In this section, we characterize the hardness of our secure convex optimization problem by proving the lower bounds for secure query complexity $T_{\mathcal{P}}(\epsilon, \delta, \epsilon^{\mathrm{adv}}, \delta^{\mathrm{adv}})$. We first present a general approach for characterizing the lower bounds which may hold for most problem classes, with the results summarized in Theorem 1. We then demonstrate how to utilize this general approach to derive lower bounds for a variety of classes of problems in Section 3.2.

## 3.1 A general framework for characterizing lower bounds

Without the requirement to secure the learner's privacy, characterizing the query complexity can follow the proof techniques developed in minimax bounds literature via reducing the optimization problem into a hypothesis testing one [22, 21]. On a high-level, we can first construct a difficult problem subclass with a set of hard-to-differentiate functions. If there exists an optimization algorithm that achieves high accuracy, we can utilize the algorithm to differentiate functions in the set. Since there exist information bounds in hypothesis testing to characterize the hardness of differentiating functions, these information bounds imply the hardness of designing optimization algorithms that achieves high accuracy.

The main challenge we face is to incorporate the requirement of securing the learner's privacy in the analysis. Recall that the secure query complexity is defined with respect to all possible adversaries, and a stronger adversary makes it harder to maintain privacy. In our proof, we focus on an ostensibly weak adversary and derive our lower bounds with respect to this adversary. While this choice seems

to lead to a weaker lower bound, we demonstrate later that there is a matching upper bound for any adversary. These two results jointly imply that no other adversary can lead to stronger lower bounds, and the bound we obtain is therefore tight.

**Constructing *difficult* problem instances.** Given a problem class $\mathcal{P} = (\mathcal{X}, \mathcal{F}, \phi)$, we construct a "difficult" subclass $\mathcal{F}' = \{f_1, \ldots, f_N\} \subseteq \mathcal{F}$, such that the functions in $\mathcal{F}'$ are hard to distinguish from one another with any possible query sequence, and yet they are sufficiently different from one another so an optimizer for one of them fails to optimize other functions to the same accuracy. With this construction, any algorithm that can reach $(\epsilon, \delta)$-accuracy can be used to "differentiate" them if we treat each function as a hypothesis in hypothesis testing. We then consider a fictitious situation in which Nature uniformly selects a function in $\mathcal{F}'$, so that for every algorithm $\mathcal{A}$, we can construct a probability space $(\Omega, \mathcal{B}, \mathbb{P})$ with the following random variables: $M \in \{1, \ldots, N\}$ encodes the random choice of selected function instance in $\mathcal{F}'$; $X^T \in \mathcal{X}^T$ are the queries issued by $\mathcal{A}$ and $\widehat{X}_T \in \mathcal{X}$ is the candidate optimizer[4]; $Y^T \in \mathcal{Y}^T$ are the responses of $\phi$ to the queries issued by $\mathcal{A}$. The way we construct such $\mathcal{F}'$ is via a "packing set" of the convex domain $\mathcal{X}$.

Suppose given a problem class $\mathcal{P} = (\mathcal{X}, \mathcal{F}, \phi)$, to set up our analysis, given a type of error measure, we first endow the instance space $\mathcal{F}$ with a *distance measure* $\pi(\cdot, \cdot)$ that has the following property: For any $x \in \mathcal{X}$ and any $\epsilon > 0$, and two functions $f, f' \in \mathcal{F}'$, we have

$$\pi(f, f') \geq 2\epsilon \text{ and } \texttt{err}(\widehat{X}_T, f) < \epsilon \implies \texttt{err}(\widehat{X}_T, f') \geq \epsilon. \tag{3}$$

In other words, an $\epsilon$-optimizer (whose estimate error with respect to the optima is no larger than $\epsilon$) of a function cannot simultaneously be an $\epsilon$-optimizer of another distinct function. It is easy to construct such distance $\pi$ satisfying (3) for any particular class $\mathcal{F}$ of continuous functions, and the design of $\pi$ usually depends on the choice of error measure. For a general $\mathcal{F}$ and function error, we can design $\pi$ over $\mathcal{F}'$ in the following way: $\pi(f, f') = \inf_{x \in \mathcal{X}} \left[ f(x) - \inf_x f(x) + f'(x) - \inf_x f'(x) \right]$, $\forall f, f' \in \mathcal{F}'$. While for point error, we can simply set $\pi(f, f') = \|x_f^* - x_{f'}^*\|$. In the following discussion, we will often implicitly restrict our discussion to a *subclass* of $\mathcal{F}$ and define an appropriate $\pi$ on that subclass based on the error measure.

Note that at the beginning, Nature will select a function $f_M$ from $\mathcal{F}'$ uniformly at random to be optimized. If one can construct such $\mathcal{F}'$ that satisfies the property specified in Eqn. (3) for a distance measure $\pi$, then we are able to show that if any learner's strategy $\mathcal{A}$ achieves a low optimization error over the class $\mathcal{F}'$, then one can use its output to construct an "estimator" $\widehat{M}_T$ that returns the true $M$ of $f_M$ with high probability. So the learner's optimization problem can be reduced to a canonical hypothesis testing problem. We formally prove this after we take into account the requirement of securing the learner's privacy.

**Adversary's estimation.** We focus on the following class of adversary who will use *proportional-sampling estimators* [20, 17] to infer the optimal point the learner is targeting, where $\widehat{X}^{\text{adv}}$ is sampled from all the queries proportionally. While incorporating a stronger adversary could lead to weaker lower bounds, as we demonstrate later, the lower bound we obtain is actually tight, as it matches the upper bound. In particular, given an observed query sequence $X^T$, the proportional-sampling estimator is defined as $\widehat{X}^{\text{adv}} = X_t$, where $t \sim \mathsf{Unif}\{1, \ldots, T\}$. Another way to define proportional-sampling estimator is as follows: The adversary first identifies a $2r$-packing set $\{\theta_1, \ldots, \theta_K\}$ over $\mathcal{X}$ (where $r = \epsilon^{\text{adv}}/L$ for function error and $r = \epsilon^{\text{adv}}$ for point error). For each $k \in [K]$, let $\mathbb{B}(\theta_k, r) = \{x \in \mathcal{X} : \|x - \theta_k\| \leq r\}$ be the $\ell_2$-norm ball with the radius of $r$ centering in $\theta_k$. Then depends on the error measure, the proportional-sampling estimator $\widehat{X}^{\text{adv}}$ can also be defined as:

$$\mathbb{P}\left(\widehat{X}^{\text{adv}} = \theta_k\right) = \frac{\sum_{t=1}^T \mathbf{1}_{\{X_t \in \mathbb{B}(\theta_k, r)\}}}{T}, \quad k = 1, \ldots, K, \tag{4}$$

where $\mathbf{1}_{\{\mathcal{E}\}}$ is the indicator function of event $\mathcal{E}$. We note that these two methods can coincide with each other when we adopt them to prove the complexity (see the proof of Lemma 1).

**Information-theoretical derivations.** We now show how to reduce the learner's optimization problem to a canonical hypothesis testing problem, taking into account of securing the learner's privacy. Though our discussions focus on function error, all analysis can be easily adapted to point error. When the context is clear, we suppress $r$ in the notation $\mathbb{B}(\theta_k, r)$ and write it as $\mathbb{B}(\theta_k)$.

Recall that our first step is to construct a subclass of functions $\mathcal{F}' \subseteq \mathcal{F}$ that we use to derive lower bounds. And then, an uniformly selected function $f \in \mathcal{F}'$ is chosen by Nature, and this $f$ will be the learner's unknown objective function. With the adversary's proportional-sampling estimator, the randomness structure leads us to build connections between the adversary's correct estimation probability and the query complexity that we are interested in quantifying. This is summarized in the following lemma.

**Lemma 1.** *Define the event $\xi_k = \{x_f^* \in \mathbb{B}(\theta_k)\}$. If the adversary follows the proportional-sampling estimator, including the one defined in* (4)*, then to ensure an algorithm is $(\epsilon^{\mathrm{adv}}, \delta^{\mathrm{adv}})$-private, we must have*

$$T_{\mathcal{P}} \geq \frac{1}{\delta^{\mathrm{adv}}} \sum_{t=1}^{T} \mathbb{P}\big(X_t \in \mathbb{B}(\theta_k) \mid \xi_k\big). \tag{5}$$

The above lemma implies that, if we can obtain the lower bound on the right hand side of the above inequality (5), we obtain the lower bound of $T$, the secure query complexity. In the discussion below, we show that conditional on the event $\xi_k$, if an algorithm achieves a low minimax error over $\mathcal{F}'$, then one can use its output to construct an estimator $\widehat{M}_T$ that returns the true $M$ most of the time.

**Lemma 2.** *Suppose an algorithm $\mathcal{A}$ attains a minimax error: $\sup_{f \in \mathcal{F}} \mathbb{P}(\mathbf{err}(\widehat{X}_T, f) \geq \epsilon) \leq \delta$. Let $\mathcal{F}' \subseteq \mathcal{F}$ be a finite set $\{f_1, \ldots, f_N\}$ such that every two distinct functions in $\mathcal{F}'$ satisfy* (3)*. Suppose $f_M$ is chosen uniformly at random from $\mathcal{F}'$, and algorithm $\mathcal{A}$ then operates with $f_M$. Then one can construct $\widehat{M}_T$ for $M$ such that the following holds:*

$$I\left(M; \widehat{M}_T \mid \xi_k\right) \geq (1 - \delta) \log |\mathcal{F}'(\theta_k)| - \log 2 > 0, \tag{6}$$

*where $I(\cdot \mid)$ represents the conditional mutual information and $\mathcal{F}'(\theta_k) = \{f_m : x_{f_m}^* \in \mathbb{B}(\theta_k), m \in [N]\}$ denotes the set of functions whose optimizers locate within the ball $\mathbb{B}(\theta_k)$ for a fix $k \in [K]$.*

Note that the above mutual information is conditional on the event $\xi_k$ and the inequality holds for every $k \in [K]$. This leads to a critical difference between the above lower bound of mutual information, in which we restrict the number of possible values of $\widehat{M}_T$ to be $|\mathcal{F}'(\theta_k)|$, comparing to that of the non-private one (which should be $N$). We have thus shown that having a low minimax optimization error over $\mathcal{F}'$ implies that the functions in $\mathcal{F}'$ can be identified most of the time. The above inequality implies that any "good" algorithm of the learner (runs for $T$ steps) should obtain non-trivial amount of information about $M$ at the end of its operation.

On the other hand, the amount of information $I(M; \widehat{M}_T \mid \xi_k)$ is well upper bounded:

**Lemma 3.** *Fix $k \in [K]$ and for any estimator $\widehat{M}_T : \mathcal{X}^T \times \mathcal{Y}^T \to \{1, \ldots, N\}$, the conditional mutual information can be upper bounded by a summation of two parts:*

$$I(M; \widehat{M}_T \mid \xi_k) \leq \sum_{t=1}^{T} \big(\mathbb{P}\left(X_t \in \mathbb{B}(\theta_k) \mid \xi_k\right) G(X_t \in \mathbb{B}(\theta_k), \xi_k) +$$
$$\mathbb{P}\left(X_t \notin \mathbb{B}(\theta_k) \mid \xi_k\right) G(X_t \notin \mathbb{B}(\theta_k), \xi_k)\big), \tag{7}$$

*where we have $G(X_t \in \mathbb{B}(\theta_k), \xi_k) = \mathbb{E}_M \mathbb{E}_{M'} D_{\mathrm{KL}}\big(\mathbb{P}(Y_t \mid M, X_t \in \mathbb{B}(\theta_k), \xi_k) \| \mathbb{P}(Y_t \mid M', X_t \in \mathbb{B}(\theta_k), \xi_k)\big)$ and $G(X_t \notin \mathbb{B}(\theta_k), \xi_k) = \mathbb{E}_M \mathbb{E}_{M'} D_{\mathrm{KL}}\big(\mathbb{P}(Y_t \mid M, X_t \notin \mathbb{B}(\theta_k), \xi_k) \| \mathbb{P}(Y_t \mid M', X_t \notin \mathbb{B}(\theta_k), \xi_k)\big)$. The expectation $\mathbb{E}_M$ (or $\mathbb{E}_{M'}$) is taken over $f_M$ (or $f_{M'}$) which is uniformly distributed over $\mathcal{F}'(\theta_k)$. And $D_{\mathrm{KL}}(\mathbb{P} \| \mathbb{Q})$ denotes the Kullback-Leibler (KL) divergence between $\mathbb{P}$ and $\mathbb{Q}$.*

The proof is provided in Appendix A.3. The above lemma characterizes the upper bound of our conditional mutual information via two parts: The first part is the cumulative correct querying

probability, while the second one is cumulative incorrect querying probability. Note that in a statistical sense, the divergence $D_{\mathrm{KL}}(\mathbb{P}(Y \mid M, X, \xi_k) \| \mathbb{P}(Y \mid M', X, \xi_k))$ quantifies how close the oracle's responses are for a given query point $x \in \mathcal{X}$ and a given pair $f_M, f_{M'}$ in $\mathcal{F}'(\theta_k)$.

Combining all pieces, we can obtain following general bound which holds for most problem classes.

**Theorem 1.** *Fix a problem class $\mathcal{P} = (\mathcal{X}, \mathcal{F}, \phi)$ and given an error measure, let $\{\theta_1, \ldots, \theta_K\}$ be a $2r$-packing set over $\mathcal{X}$ (where $r = \epsilon^{\mathrm{adv}}/L$ for function error and $r = \epsilon^{\mathrm{adv}}$ for point error). Suppose there exists a function subclass $\mathcal{F}' \subseteq \mathcal{F}$ such that it satisfies the following conditions:*

1. *the distance measure $\pi$ defined in Eqn. (3) holds for any two distinct functions $f, f' \in \mathcal{F}'$;*

2. *for some $C^* > 0$, $G(x \in \mathbb{B}(\theta_k), \xi_k) \leq C^*, \forall x \in \mathbb{B}(\theta_k)$ and $f_M, f_{M'} \in \mathcal{F}'(\theta_k)$;*

3. *$G(x \notin \mathbb{B}(\theta_k), \xi_k) = 0, \forall x \in \mathbb{B}(\theta_k)$ and $f_M, f_{M'} \in \mathcal{F}'(\theta_k)$.*

*Then the secure query complexity satisfies: $T_{\mathcal{P}} \geq \Omega\left(\frac{1-\delta}{C^* \delta^{\mathrm{adv}}} \log |\mathcal{F}'(\theta_k)|\right)$.*

**Remark 2.** The above general lower bound is a direct result of applying Lemma 1 to Lemma 3. Though this lower bound holds generally, it is only tight for certain problem classes. The third condition also provides a hint on how to construct function subclass: Given a coarsening adversary's estimation ball, the functions whose optimizer lie within this ball should be *indistinguishable* based on the function value and gradient information calculated outside this ball.

## 3.2 Deriving lower bounds

In the section, we demonstrate how to utilize the above analysis for different problem classes. Note that from Theorem 1, the derivation of the lower bounds reduces to finding the problem subclass that satisfies the three listed conditions.

**(Noisy) binary search** We first explore the secure query complexity of secure binary search $\mathcal{P} = \{[0, 1], \mathcal{F}^{\mathrm{Abs}}, \phi^{\mathsf{sign}}\}$ and secure noisy binary search $\mathcal{P} = \{[0, 1], \mathcal{F}^{\mathrm{Abs}}, \phi^{\mathsf{sign}, p}\}$ as defined in Section 2.1. The result of secure binary search can be summarized as follows.

**Theorem 2** (Secure Binary Search). *Given small $\delta, \delta^{\mathrm{adv}} \in (0, 1)$, and $2\epsilon \leq \epsilon^{\mathrm{adv}} \leq \delta^{\mathrm{adv}}/2$,[5] for binary search $\mathcal{P} = \{[0, 1], \mathcal{F}^{\mathrm{Abs}}, \phi^{\mathsf{sign}}\}$, the secure query complexity is lower bounded as: $T_{\mathcal{P}} \geq \Omega\left(\frac{1-\delta}{\delta^{\mathrm{adv}}} \log(\epsilon^{\mathrm{adv}}/\epsilon)\right).$*

The full proof of the above theorem is in Appendix B.1.

**Remark 3.** We obtain the same lower bound as in prior works on secure binary search in the Bayesian setting [20, 17, 19], where a lower bound in the order of $\Omega(\log(\epsilon^{\mathrm{adv}}/\epsilon)/\delta^{\mathrm{adv}})$ was derived. Our use of a different technique based on the minimax analysis allows us to generalize the results to noisy binary search, in which the oracle response is correct with probability $p$.

**Theorem 3** (Secure Noisy Binary Search). *Given small $\delta, \delta^{\mathrm{adv}} \in (0, 1)$, and $2\epsilon \leq \epsilon^{\mathrm{adv}} \leq \delta^{\mathrm{adv}}/2$, for secure noisy binary search $\mathcal{P} = \{[0, 1], \mathcal{F}^{\mathrm{Abs}}, \phi^{\mathsf{sign}, p}\}$, where $p \in (1/2, 1)$, the secure query complexity is lower bounded as: $T_{\mathcal{P}} \geq \Omega\left(\frac{1-\delta}{\delta^{\mathrm{adv}} c(p)} \log(\epsilon^{\mathrm{adv}}/\epsilon)\right)$, where $c(p) > 0$ is a constant value depending only on the parameter $p$.*

We defer the detailed proof to Appendix B.2. We obtain a similar bound to the work by [19] for noisy binary search, while their bound contains more refined constants.

**Remark 4.** For (non-secure) noisy binary search, it is shown [18] that the lower bound of convergence rate is $\mathbb{E}[|x^* - X_T|] = o(c_1^{-T})$ for some constant $c_1 > 1$ depending only on $p$. Our secure variant converges at the order of $\epsilon^{\mathrm{adv}} c_2^{-T}$, where $c_2$ is a fixed constant depending on $p, \delta^{\mathrm{adv}}$. This is tight up to a multiplicative constant compared with the classic result.

**Stochastic convex optimization.** We now present our main results for secure stochastic convex optimization. We state our private complexity results with restricting $\mathcal{X}$ to be $[0,1]^d$. Recall that $\mathcal{F}\kappa$ is the set of $\kappa$-uniformly convex functions.

**Theorem 4** (Secure Stochastic Convex Optimization). *Consider the problem class $\mathcal{P} = [0,1]^d, \mathcal{F}^\kappa, \phi^{(1)}$) with a stochastic first-order oracle $\phi^{(1)}$. Then for any $2\sqrt{d}\epsilon \leq \epsilon^{\mathrm{adv}} \leq (\delta^{\mathrm{adv}})^{1/d}$, small $\delta, \delta^{\mathrm{adv}} \in (0,1)$, the following secure query complexity holds: $T_{\mathcal{P}} \geq \Omega\left(\frac{\sigma^2(\log 2 - h_2(\delta))}{\delta^{\mathrm{adv}}\epsilon^{(2\kappa-2)/\kappa}}\right)$ for function error, $T_{\mathcal{P}} \geq \Omega\left(\frac{\sigma^2(\log 2 - h_2(\delta))}{\delta^{\mathrm{adv}}\epsilon^{2\kappa-2}}\right)$ for point error.*

We defer the proof to Appendix B.3. The key step is to construct a "difficult" function subclass $\mathcal{F}'$ so that the functions in $\mathcal{F}'$ are *indistinguishable* based only the function and gradient information when the query points are outside adversary's estimation region (Condition (3) in Theorem 1).To achieve this, we start with some base convex functions. We then construct the function $f$ in $\mathcal{F}'$ via a maximum operator. This construction helps us ensure the third condition in Theorem 1 is satisfied. An example of the construction when $\kappa = 2$ is given in Fig 1.

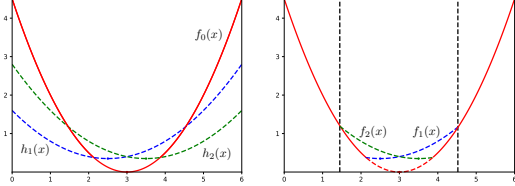

Figure 1: Left: Base functions $f_0(x) = 0.5|x - 3|^2$, $h_1(x) = 0.2|x - (3 - 0.5)|^2 - 1.6$ and $h_2(x) = 0.2|x - (3 + 0.5)|^2 - 1.6$. Right: $f_1(x) = \max\{f_0(x), h_1(x)\}$ and $f_2(x) = \max\{f_0(x), h_2(x)\}$.

We offer a few observations of our results. First, our results match the lower bounds in non-secure convex optimization. In particular, when $\kappa = 2$ (i.e., strongly convex functions), our lower bound matches the known lower bound of standard convex optimization, $\Omega(1/T)$ (because $T_{\mathcal{P}} \geq \Omega(1/\epsilon)$) for function error and $\Omega(1/\sqrt{T})$ for point error. As $\kappa \to \infty$ (i.e., non-strongly convex functions), our lower bound, in the order of $\Omega(1/\sqrt{T})$ for function error, also matches the classic result for Lipschitz convex function optimization. The convergence for point error would fail with non-strongly convex functions - this corresponds to the worst case Lipschitz convex functions. As an illustration, it is pointless to "converge" to a single optima for a flat line, a non-strongly convex function.

Second, our privacy constraint leads to a multiplicative penalty of $1/\delta^{\mathrm{adv}}$ in both error measure. This can be considered as a complexity price to pay for the increased privacy. Intuitively, one can also view this penalty as the learner trying to fool the adversary by hiding her non-secure learning strategy within other $\Theta(1/\delta^{\mathrm{adv}})$ fictitiously designed identical strategies.

Third, while our bounds do not seem to explicitly depend on $\epsilon^{\mathrm{adv}}$, it is hidden in the logarithmic factor. To be more concrete, according to our Lemma 2, $\epsilon^{\mathrm{adv}}$ will impact the value of $|\mathcal{F}'(\theta_k)|$, which is bounded by $\Omega(\epsilon^{\mathrm{adv}}/\epsilon)$. After taking the logarithm to get $\Omega(\log(\epsilon^{\mathrm{adv}}/\epsilon))$, we conclude that this term is dominated by $\Omega(1/\epsilon^\alpha)$ for any $\alpha \geq 1$.

Finally, our results can be extended to the settings with general noisy oracles. As long as Gaussian noise is a subclass of the noise distribution, our lower bounds hold. The Gaussian assumption serves the goal for proving the lower bounds. In the following section, our algorithm and upper bound analysis will also go through for all sub-Gaussian noise oracles. For the ease of presentation, we will focus on Gaussian noise model for the current paper.

## 4   An Optimal Secure Optimization Strategy

We present a simple and intuitive algorithm that is *optimal* in the sense that it obtains the matching upper bounds in secure query complexity when an arbitrary adversary can present. To secure the learner's privacy, imagine that if the learner performs query uniformly at random for each time step, while the learner sacrifices the learning efficiency, the privacy is secured as no adversary can infer anything from where the leaner queries. The high-level intuition of our algorithm is to mix (secure but non-efficient) uniform query protocol and (efficient but non-secure) standard methods from the optimization literature.

To simplify the presentation, we focus on the one-dimensional case with domain $\mathcal{X} \in [0, 1]$. To be consistent with standard convex optimization algorithms, we present our secure learning protocol where the objective is to optimize the estimation error rate. Inspired by the replicated bisection strategy proposed by Tsitsiklis et al. [17], the general idea of the protocol is as follows: Fixed an oracle budget $T$, we divide this budget into $\lfloor T/S \rfloor$ phases over each of $S = \lfloor 1/\delta^{\text{adv}} \rfloor$ many queries. We also divide the domain $[0, 1]$ into equal sub-intervals with length of $\delta^{\text{adv}}$. Within each phase, the learner *symmetrically* submits one query to each sub-interval. Among these queries in each phase, there is one query that is consecutively updated according to learner's confidential computation oracle, which can be any efficient algorithm for stochastic convex optimization. The layer of randomization over $\lfloor 1/\delta^{\text{adv}} \rfloor$-length intervals is the key device to secure the learner's privacy.[6] The details of our secure learning protocol, and together with an example of computation oracle, are included in Appendix B.4.

Below is the formal statement that this secure learning protocol leads to a upper bound of secure query complexity that matches our lower bound up to a logarithmic factor. The proof is in Appendix B.4.

**Theorem 5.** *Fix $\epsilon^{\text{adv}}, \delta^{\text{adv}} \in (0, 1)$ such that $2\epsilon^{\text{adv}} < \delta^{\text{adv}}$. Learning protocol detailed in Algorithm 1, together with the computation oracle detailed in Algorithm 2, can return an estimator $\widehat{X}_T$ for the learner such that for any $f \in \mathcal{F}^\kappa, \kappa > 1$, $f(\widehat{X}_T) - f^* \leq \tilde{\mathcal{O}}\big((T\delta^{\text{adv}})^{-\frac{\kappa}{2\kappa-2}}\big)$ and $|\widehat{X}_T - X^*| \leq \tilde{\mathcal{O}}\big((T\delta^{\text{adv}})^{-\frac{1}{2\kappa-2}}\big)$ hold with probability at least $1 - \delta$. Furthermore, the queries generated from such learning protocol are $(\epsilon^{\text{adv}}, \delta^{\text{adv}})$-private.*

**Remark 5.** The upper bound holds w.r.t. *arbitrary* adversary strategies. It is easy to verify that the above convergence rate can be translated to an upper bound that matches the lower bound of query complexity. Therefore, our lower bound derived via assuming a specific type of adversary is tight.

# 5   Discussions and Future Directions

This work studies the secure stochastic convex optimization problem. We present a general information-theoretical analysis and characterize lower bounds. We also give an efficient secure learning protocol with matching upper bounds. A number of open questions remain. In particular, while our current results work for high-dimensional problem instances, we have not analyzed the secure query complexity's dependence on the input dimensions. Characterizing this dependency would be an interesting future direction. In addition, although our lower bound is tight, it relies on assuming the proportional-sampling adversarial strategy. It is unclear whether we can generalize our analysis when considering other certain types of adversaries.

## Broader Impact

In this work, we explore the problem of securing the privacy of the learner against a spying adversary. In a broader context, we explore the limit of securing the decision maker's unobservable intent/goal when the query decisions to achieve the intent/goal are observable. Our results, while being theoretical in nature, have potential impacts in providing instructions for designing better security tools to ensure that people's online activities do not create unintended leakage of private information. On the other hand, the discussion on the adversarial strategies could also lead to more delicate attacks, especially to those who are not aware of the existence of attacks from potential adversaries.

## Acknowledgments and Disclosure of Funding

We thank Kuang Xu, Jiaming Xu and Dana Yang for the helpful discussions and the anonymous reviewers for their valuable comments and suggestions. This work is supported in part by ONR Grant N00014-20-1-2240.

## Footnotes

[1]In this paper, we use "she" to address the learner and "he" to address the adversary. In addition, we denote our problem as *secure* optimization instead of *private* optimization to differentiate with the works in differential privacy. Generally speaking, the goal of differential privacy is to protect the privacy of individual data contributors, while our goal is to secure the privacy of the learner.

[2]We use superscript $\mathrm{adv}$ for the privacy notion since it is related to the $\mathrm{adversary}$'s estimation.

[3]The dependency on $\epsilon^{\mathrm{adv}}$ is in the logarithmic factor.

[4]We sometimes use $\widehat{X}_T$ instead of $\widehat{X}$ to emphasize its dependency on $T$.

[5]We restrict the parameter range to exclude trivial cases. For example, if $\epsilon^{\mathrm{adv}} > \delta^{\mathrm{adv}}/2$, the privacy requirement is too strong to be achieved. Consider a naive adversary that obtains an estimate by drawing a point uniformly at random in $[0, 1]$. In this case, with probability greater than $\delta^{\mathrm{adv}}$, the adversary's estimate is within $\delta^{\mathrm{adv}}/2$ to the optima (due to uniform sampling). If $\epsilon^{\mathrm{adv}} > \delta^{\mathrm{adv}}/2$, the privacy requirement is violated.

[6]Randomization here means that the adversary can't do better by guessing uniformly at random.

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
