[Supplementary Material]

# A Missing Proofs

## A.1 Proof of Lemma 1

*Proof.* For any querying strategy that is $(\epsilon^{\text{adv}}, \delta^{\text{adv}})$-private, it must satisfy $\mathbb{P}(\text{err}(\widehat{X}^{\text{adv}}, f) \leq \epsilon^{\text{adv}}) \leq \delta^{\text{adv}}$. We choose function error to prove this lemma. Suppose the adversary's estimator $\widehat{X}^{\text{adv}}$ is obtained through the proportional-sampling, then we have

$$\mathbb{P}\left(\text{err}(\widehat{X}^{\text{adv}}, f) \leq \epsilon^{\text{adv}}\right) = \sum_{t=1}^{T} \mathbb{P}\left(\widehat{X}^{\text{adv}} = X_t\right) \mathbb{P}\left(\text{err}(X_t, f) \leq \epsilon^{\text{adv}} | \widehat{X}^{\text{adv}} = X_t\right)$$
$$\geq \frac{\sum_{t=1}^{T} \mathbb{P}\left(\|X_t - x^*\| \leq \epsilon^{\text{adv}}/L\right)}{T}, \tag{8}$$

where $L$ is the Lipschitz constant of function $f$. To ensure $(\epsilon^{\text{adv}}, \delta^{\text{adv}})$-privacy, it deduces that

$$T \geq \frac{1}{\delta^{\text{adv}}} \left( \sum_{t=1}^{T} \mathbb{P}\left(\|X_t - x^*\| \leq \epsilon^{\text{adv}}/L\right) \right). \tag{9}$$

Furthermore, note that $f$ is uniformly-distributed among $\mathcal{F}'$, we have following

$$\sum_{t=1}^{T} \mathbb{P}\left(\|X_t - x^*\| \leq \epsilon^{\text{adv}}/L\right) = \sum_{t=1}^{T} \sum_{k \in [K]} \mathbb{P}\left(\|X_t - x^*\| \leq \epsilon^{\text{adv}}/L \mid \xi_k\right) \cdot \mathbb{P}\left(\xi_k\right)$$
$$= \sum_{t=1}^{T} \mathbb{P}\left(X_t \in \mathbb{B}(\theta_k, \epsilon^{\text{adv}}/L) \mid \xi_k\right).$$

This proves the lemma.

For adversary's strategy defined in (4), the proof is slightly different and we include it below for completeness.

For each $k \in [K]$, let $\Gamma_k = \{X_t : X_t \in \mathbb{B}(\theta_k)\}_{t \geq 1}$ denote the set of queries that lie within the ball $\mathbb{B}(\theta_k, \epsilon^{\text{adv}}/L)$. For the adversary's estimator defined in (4), we also have following reduction to the adversary's probability of correct estimation:

$$\mathbb{P}\left(\text{err}(\widehat{X}^{\text{adv}}, f) \leq \epsilon^{\text{adv}} | x_f^* \in \mathbb{B}(\theta_k, \epsilon^{\text{adv}}/L)\right) \geq \mathbb{P}\left(\|\widehat{X}^{\text{adv}} - x_f^*\| \leq \epsilon^{\text{adv}}/L | x_f^* \in \mathbb{B}(\theta_k, \epsilon^{\text{adv}}/L)\right)$$
$$= \mathbb{P}\left(\widehat{X}^{\text{adv}} = \theta_k | x_f^* \in \mathbb{B}(\theta_k, \epsilon^{\text{adv}}/L)\right)$$
$$= \mathbb{E}\left(\frac{|\Gamma_k|}{\sum_k |\Gamma_k|} \mid x_f^* \in \mathbb{B}(\theta_k, \epsilon^{\text{adv}}/L)\right)$$
$$= \frac{\mathbb{E}\left(|\Gamma_k| \mid x_f^* \in \mathbb{B}(\theta_k, \epsilon^{\text{adv}}/L)\right)}{T}.$$

Note that $|\Gamma_k| = \sum_{t=1} \mathbf{1}_{\{X_t \in \mathbb{B}(\theta_k)\}}$. Thus, we have that

$$\mathbb{P}\left(\text{err}(\widehat{X}^{\text{adv}}, f) \leq \epsilon^{\text{adv}} | x_f^* \in \mathbb{B}(\theta_k, \epsilon^{\text{adv}}/L)\right) \geq \frac{\sum_{t=1} \mathbb{P}\left(X_t \in \mathbb{B}(\theta_k, \epsilon^{\text{adv}}/L) \mid \xi_k\right)}{T}.$$

This is the desired result in the lemma.

For point error, the above analysis can be easily carried over by adjusting the term $\epsilon^{\text{adv}}/L$ to $\epsilon^{\text{adv}}$. $\square$

## A.2 Proof of Lemma 2

*Proof.* We note that conditional on event $\xi_k$, such estimator $\widehat{M}_T$ can be defined as

$$\widehat{M}_T(X^T, Y^T) = \underset{m: f_m \in \mathcal{F}'(\theta_k)}{\text{argmin}} \text{err}(\widehat{X}_T, f_m), \tag{10}$$

which simply predicts the function in $\mathcal{F}'$ for which the error of $\widehat{X}_T$ is the smallest. Since $\widehat{X}_T$ is $\sigma(X^T, Y^T)$-measurable, the estimator $\widehat{M}_T$ is indeed a function only of the information available to $\mathcal{A}$ after time $T$. We define, for each $f_i \in \mathcal{F}'(\theta_k)$, the event

$$\mathcal{E}_i \triangleq \left\{ \texttt{err}(\widehat{X}_T, f_i) \geq \epsilon \right\}.$$

Indeed, if $\mathcal{E}_i$ does not occur, then from the fact that $\pi(f_i, f_j) \geq 2\epsilon$ for all $j \neq i$ and from (3) we deduce that

$$\texttt{err}(\widehat{X}_T, f_j) > \epsilon > \texttt{err}(\widehat{X}_T, f_i), \qquad \forall j \neq i.$$

So it must be the case that $\widehat{M}_T = i$. Therefore,

$$
\begin{aligned}
\delta &\geq \max_{f_i \in \mathcal{F}'(\theta_k)} \mathbb{P}(\mathcal{E}_i | M = i, \xi_k) \\
&\geq \max_{f_i \in \mathcal{F}'(\theta_k)} \mathbb{P}(\widehat{M}_T \neq i | M = i, \xi_k) \geq \mathbb{P}(\widehat{M}_T \neq M | \xi_k).
\end{aligned}
$$

In addition, we note that

$$
\begin{aligned}
\mathbb{P}\left( \texttt{err}(\widehat{X}_T, f) \geq \epsilon \right) &= \sum_k \mathbb{P}\left( \texttt{err}(\widehat{X}_T, f) \geq \epsilon \mid \xi_k \right) \cdot \mathbb{P}(\xi_k) \\
&= \mathbb{P}\left( \texttt{err}(\widehat{X}_T, f) \geq \epsilon \mid \xi_k \right).
\end{aligned}
$$

Thus, we have $\mathbb{P}(\widehat{M}_T \neq M \mid \xi_k) \leq \delta$. Then by Fano's inequality,

$$\delta \geq \mathbb{P}\left( \widehat{M}_T \neq M \mid \xi_k \right) \geq 1 - \frac{I\left( M; \widehat{M}_T \mid \xi_k \right) + \log 2}{\log |\mathcal{F}'(\theta_k)|}.$$

Rearranging the above inequality will yield us desired result. $\qquad\square$

## A.3 Proof of Lemma 3

*Proof.* Our proof is similar to the information radius bound established in [11] where the crux difference is that we mainly operate with the information that is additionally conditional on the event $\xi_k$. First, note that by chain rule of conditional mutual information, we have

$$I\left( M; \widehat{M} \mid \xi_k \right) \leq I\left( M; X^T, Y^T \mid \xi_k \right) \tag{11}$$

$$= \sum_{t=1}^T I\left( M; X_t, Y_t \mid X^{t-1}, Y^{t-1}, \xi_k \right) \tag{12}$$

$$= \sum_{t=1}^T I\left( M; X_t \mid X^{t-1}, Y^{t-1}, \xi_k \right) + \sum_{t=1}^T I\left( M; Y_t \mid X^t, Y^{t-1}, \xi_k \right) \tag{13}$$

$$= \sum_{t=1}^T I\left( M; Y_t \mid X^t, Y^{t-1}, \xi_k \right), \tag{14}$$

where (11) is due to the data processing inequality, (12) and (13) are the chain rule of conditional mutual information, and the last equality (14) is the reason that the choice of $X_t$ is independent of $M$ given the information $(X^{t-1}, Y^{t-1})$.

Note that for a random triple $(X_1, X_2, X_3) \in \mathcal{X}_1 \times \mathcal{X}_2 \times \mathcal{X}_3$, if $X_2$ and $X_3$ are conditionally independent given $X_1$ given $\mathbb{P}$, then the conditional mutual information between $X_2$ and $X_3$ given $X_1$ is defined as:

$$
\begin{aligned}
I(X_2; X_3 \mid X_1) &= D_{\mathrm{KL}}\left( \mathbb{P}(X_2, X_3 \mid X_1) \| \mathbb{P}(X_2 \mid X_1) \times \mathbb{P}(X_3 \mid X_1) \mid \mathbb{P}(X_1) \right) \tag{15} \\
&= D_{\mathrm{KL}}\left( \mathbb{P}(X_3 \mid X_1, X_2) \| \mathbb{P}(X_3 \mid X_1) \mid \mathbb{P}(X_1, X_2) \right) \tag{16}
\end{aligned}
$$

where (16) is due to the Bayes' rule. Observe that $Y_t$ and $M$ are conditionally independent given the information $(X^t, Y^{t-1}, \xi_k)$, in other words, $M \to (X^t, Y^{t-1}, \xi_k) \to Y_t$ is a Markov chain. Thus, fix some $t$ and consider the conditional mutual information we obtain in (14),

$$
\begin{aligned}
&I\left(M; Y_t \mid X^t, Y^{t-1}, \xi_k\right) \\
&= D_{\mathrm{KL}}\left(\mathbb{P}(Y_t \mid M, X^t, Y^{t-1}, \xi_k) \| \mathbb{P}(Y^t \mid X^t, Y^{t-1}, \xi_k) \mid \mathbb{P}(M, X^t, Y^{t-1}, \xi_k)\right),
\end{aligned} \tag{17}
$$

For any estimator $\widehat{M} : X^T \times Y^T \to \{1, \ldots, N\}$, and any sequence of conditional probability measures $\{\mathbb{Q}(Y_t | X^t, Y^{t-1})\}_{t=1}^T$ on $\{\Omega, \mathcal{B}\}$ that satisfying following conditions:

$$
\mathbb{P}\left(Y_t \mid X^t, Y^{t-1}\right) \ll \mathbb{Q}\left(Y_t \mid X^t, Y^{t-1}\right), \forall t \in [T], \tag{18}
$$

where $\mathbb{P} \ll \mathbb{Q}$ implies that $\mathbb{P}$ is absolute continuous w.r.t. $\mathbb{Q}$. Note that by definition of conditional mutual information, we can write the (17) as follows:

$$
\begin{aligned}
(17) &= \mathbb{E}\left[\log \frac{d\mathbb{P}(Y_t \mid M, X^t, Y^{t-1}, \xi_k)}{d\mathbb{P}(Y_t \mid X^t, Y^{t-1}, \xi_k)}\right] \\
&= \mathbb{E}\left[\log \frac{d\mathbb{P}(Y_t \mid M, X^t, Y^{t-1}, \xi_k)}{d\mathbb{Q}(Y_t \mid X^t, Y^{t-1}, \xi_k)}\right] - \mathbb{E}\left[\log \frac{d\mathbb{P}(Y_t \mid X^t, Y^{t-1}, \xi_k)}{d\mathbb{Q}(Y_t \mid X^t, Y^{t-1}, \xi_k)}\right] \tag{19} \\
&= D_{\mathrm{KL}}\left(\mathbb{P}(Y_t \mid M, X^t, Y^{t-1}, \xi_k) \| \mathbb{Q}(Y^t \mid X^t, Y^{t-1}, \xi_k) \mid \mathbb{P}(M, X^t, Y^{t-1}, \xi_k)\right) - \\
&\qquad D_{\mathrm{KL}}\left(\mathbb{P}(Y_t \mid X^t, Y^{t-1}, \xi_k) \| \mathbb{Q}(Y^t \mid X^t, Y^{t-1}, \xi_k) \mid \mathbb{P}(X^t, Y^{t-1}, \xi_k)\right) \tag{20} \\
&\leq D_{\mathrm{KL}}\left(\mathbb{P}(Y_t \mid M, X^t, Y^{t-1}, \xi_k) \| \mathbb{Q}(Y^t \mid X^t, Y^{t-1}, \xi_k) \mid \mathbb{P}(M, X^t, Y^{t-1}, \xi_k)\right) \tag{21}
\end{aligned}
$$

where (19) and (20) are from the condition (18), and (21) is due to the fact that the mutual information are non-negative. Taking the summation over time $t$, we obtain that:

$$
\begin{aligned}
I(M; \widehat{M} \mid \xi_k) &\leq \sum_{t=1}^T D_{\mathrm{KL}}\left(\mathbb{P}\left(Y_t \mid M, X^t, Y^{t-1}, \xi_k\right) \| \mathbb{Q}\left(Y_t \mid X^t, Y^{t-1}, \xi_k\right) \mid \mathbb{P}\left(M, X^t, Y^{t-1}, \xi_k\right)\right) \\
&= \sum_{t=1}^T D_{\mathrm{KL}}\left(\mathbb{P}(Y_t \mid M, X_t, \xi_k) \| \mathbb{Q}\left(Y_t \mid M', X_t, \xi_k\right) \mid \mathbb{P}\left(M, X^t, Y^{t-1}, \xi_k\right)\right), \tag{22}
\end{aligned}
$$

where (22) is by hypothesis on oracle's behavior: $(X^{t-1}, Y^{t-1}) \to (M, X_t) \to Y_t$ is a Markov chain. Thus, we can write $\mathbb{P}(Y_t \mid M, X^t, Y^{t-1}, \xi_k)$ as $\mathbb{P}(Y_t \mid M, X_t, \xi_k)$.

At each round $t$, take $\mathbb{Q}\left(Y_t \mid X^t, Y^{t-1}, \xi_k\right) = \mathbb{Q}\left(Y_t \mid X_t, \xi_k\right)$, and if we set $\mathbb{Q}\left(Y_t \mid X_t, \xi_k\right)$ to be $\mathbb{P}(Y_t | M, X_t, \xi_k)$ with $f_M$ uniformly distributed in $\mathcal{F}'(\theta_k)$, we will have following:

$$
\begin{aligned}
\mathbb{Q}\left(Y_t \mid X_t, \xi_k\right) &= \frac{1}{|\mathcal{F}'(\theta_k)|} \sum_{i \in \mathcal{F}'(\theta_k)} \mathbb{P}\left(Y_t \mid M = i, X_t, \xi_k\right) \\
&= \mathbb{E}_M \mathbb{P}\left(Y_t \mid M, X_t, \xi_k\right).
\end{aligned}
$$

Then, introducing an independent copy of $M$ ($M'$), and noting that $\mathbb{Q}\left(Y_t \mid X_t, \xi_k\right) = \mathbb{E}_{M'} \mathbb{P}\left(Y_t \mid M', X_t, \xi_k\right)$, we can obtain following upper bound of the conditional mutual information

we are operating on:

$$I(M; \widehat{M} \mid \xi_k) \leq \sum_{t=1}^{T} \mathbb{E}_{M'} D_{\mathrm{KL}} \left( \mathbb{P}(Y_t \mid M, X_t, \xi_k) \| \mathbb{P}(Y_t \mid M', X_t, \xi_k) \mid \mathbb{P}(M, X_t, \xi_k) \right) \tag{23}$$

$$= \sum_{t=1}^{T} \mathbb{E}_{M, X_t, \xi_k} \mathbb{E}_{M'} D_{\mathrm{KL}} \left( \mathbb{P}(Y_t \mid M, X_t, \xi_k) \| \mathbb{P}(Y_t \mid M', X_t, \xi_k) \right)$$

$$= \sum_{t=1}^{T} \sum_{M, X_t, \xi_k} \mathbb{P}(M \mid X_t, \xi_k) \, \mathbb{P}(X_t \mid \xi_k) \, \mathbb{P}(\xi_k) \, \mathbb{E}_{M'} D_{\mathrm{KL}} \left( \mathbb{P}(Y_t \mid M, X_t, \xi_k) \| \mathbb{P}(Y_t \mid M', X_t, \xi_k) \right)$$

$$= \sum_{t=1}^{T} \sum_{x \in \mathcal{X}} \mathbb{P}(X_t = x \mid \xi_k) \sum_{M, \xi_k} \mathbb{P}(M \mid X_t = x, \xi_k) \, \mathbb{P}(\xi_k) \cdot$$
$$\qquad\qquad \mathbb{E}_{M'} D_{\mathrm{KL}} \left( \mathbb{P}(Y_t \mid M, X_t = x, \xi_k) \| \mathbb{P}(Y_t \mid M', X_t = x, \xi_k) \right)$$

$$= \sum_{t=1}^{T} \Bigg( \mathbb{P}(X_t \in \mathbb{B}(\theta_k) \mid \xi_k) \, \mathbb{E}_M \mathbb{E}_{M'} D_{\mathrm{KL}} \left( \mathbb{P}(Y_t \mid M, X_t \in \mathbb{B}(\theta_k), \xi_k) \| \mathbb{P}(Y_t \mid M', X_t \in \mathbb{B}(\theta_k), \xi_k) \right) +$$

$$\mathbb{P}(X_t \notin \mathbb{B}(\theta_k) \mid \xi_k) \, \mathbb{E}_M \mathbb{E}_{M'} D_{\mathrm{KL}} \left( \mathbb{P}(Y_t \mid M, X_t \notin \mathbb{B}(\theta_k), \xi_k) \| \mathbb{P}(Y_t \mid M', X_t \notin \mathbb{B}(\theta_k), \xi_k) \right) \Bigg),$$
$$\tag{24}$$

where we have used the convexity property of the divergence and inequality (23) is then the result of Jensen's inequality. The last equality (24) used the fact that Nature selects function $f$ uniformly at random. The expectation $\mathbb{E}_M$ (or $\mathbb{E}_{M'}$) is taken over $f_M$ (or $f_{M'}$) which is uniformly distributed over $\mathcal{F}'(\theta_k)$. $\qquad\square$

# B  Proofs for main results

## B.1  Proofs for Theorem 2

*Proof.* Let $\Lambda_\epsilon = \{\alpha_1, \ldots, \alpha_N\}$ and $\Lambda_{\epsilon^{\mathrm{adv}}} = \{\theta_1, \ldots, \theta_K\}$ denote maximal $2\epsilon$-packing set, $2\epsilon^{\mathrm{adv}}$-packing set in $[0, 1]$, respectively. We define following function subclass $\mathcal{F}' = \{f_\alpha(x)\}_{\alpha \in \Lambda_\epsilon} \subset \mathcal{F}^{\mathrm{Abs}}$:

$$f_\alpha(x) = |x - \alpha|, \quad \alpha \in \Lambda_\epsilon. \tag{25}$$

It is easy to see that, $N \geq 1/\epsilon$ and $K \geq 1/\epsilon^{\mathrm{adv}}$. Furthermore, we also have $\pi(f_{\alpha_i}, f_{\alpha_j}) = |\alpha_i - \alpha_j| \geq 2\epsilon$. Now let $f_{\alpha_M}$ be the function selected by Nature among $\mathcal{F}'$, and recall that $\xi_k$ denotes the event $\{\alpha_M \in [\theta_k - \epsilon^{\mathrm{adv}}, \theta_k + \epsilon^{\mathrm{adv}}]\}$. Then, by Lemma 2, we have

$$I(M; \widehat{M}_T | \xi_k) \geq (1 - \delta) \log |\mathcal{F}'(\theta_k)| - \log 2, \tag{26}$$

where $|\mathcal{F}'(\theta_k)| = \epsilon^{\mathrm{adv}}/\epsilon$ by construction. On the other hand, we can also upper bound the above conditional mutual information. From the fact $I(X; Y | Z) = H(X|Z) - H(X|Y, Z)$ and entropy $H(\cdot)$ is nonnegative, we have $I(X; Y | Z) \leq H(X|Z)$, thus

$$I(M; \widehat{M}_T | \xi_k) \leq H(Y^T | \xi_k) \tag{27}$$

$$\leq H(Y_1 | \xi_k) + \sum_{t=1}^{T-1} H(Y_{t+1} | Y_1, \ldots, Y_t, \xi_k) \qquad \text{(By chain rule)}$$

Note that, by definition, we have

$$H(Y_{t+1} | Y_1, \ldots, Y_t, \xi_k) = \sum_{y_1, \ldots, y_t} \mathbb{P}(Y_1 = y_1, \ldots, Y_t = y_t | \xi_k) H(Y_{t+1} | Y_1 = y_1, \ldots, Y_t = y_t, \xi_k)$$
$$\tag{28}$$

Observe that, conditional on the event $\xi_k$, if an algorithm $\mathcal{A}_t(y_1, \ldots, y_t)$ outputs the next query $X_{t+1}$ which is smaller than $\theta_k - \epsilon^{\mathrm{adv}}$, then we must have $Y_{t+1} = -1$, while if it is larger than $\theta_k + \epsilon^{\mathrm{adv}}$, then we have $Y_{t+1} = +1$. Moreover when $X_{t+1}$ is in the range $[\theta_k - \epsilon^{\mathrm{adv}}, \theta_k + \epsilon^{\mathrm{adv}}]$, $H(Y_{t+1}|\cdot)$ can take only two values, namely $+1$ and $-1$. Thus, $H(Y_{t+1}|X_{t+1} \in [\theta_k - \epsilon^{\mathrm{adv}}, \theta_k + \epsilon^{\mathrm{adv}}]) \leq 1$. The above observations give us following result

$$\sum_{y_1, \ldots, y_t} \mathbb{P}(Y_1 = y_1, \ldots, Y_t = y_t | \xi_k) H(Y_{t+1} | Y_1 = y_1, \ldots, Y_t = y_t, \xi_k) \tag{29}$$

$$= \sum_{y_1, \ldots, y_t : \mathcal{A}_t(y_1, \ldots, y_t) \in \mathbb{B}(\theta_k)} \mathbb{P}(Y_1 = y_1, \ldots, Y_t = y_t | \xi_k) H(Y_{t+1} | Y_1 = y_1, \ldots, Y_t = y_t, \xi_k) \tag{30}$$

$$\leq \mathbb{P}(X_{t+1} \in \mathbb{B}(\theta_k) | \xi_k). \tag{31}$$

With inequality in (26), we conclude our result. $\qquad\square$

## B.2 Proofs for Theorem 3

*Proof.* The proof is overall similar to the one in secure binary search, we also construct two packing sets $\Lambda_\epsilon$ and $\Lambda_{\epsilon^{\mathrm{adv}}}$ to set up our analysis. The only difference is how we bound the KL divergence of two probability measures induced by two randomly selected function instances in $\mathcal{F}'$. In particular, note that conditional on event $\xi_k$, i.e., $x^* \in \mathbb{B}(\theta_k) = [\theta_k - \epsilon^{\mathrm{adv}}, \theta_k + \epsilon^{\mathrm{adv}}]$, for any function $f$ in $\mathcal{F}'(\theta_k)$, when the query $X_t$ is smaller than $\theta_k - \epsilon^{\mathrm{adv}}$, we have $\mathbb{P}(Y_{t+1} = -1) = p$ and $\mathbb{P}(Y_{t+1} = +1) = 1 - p$; while the query $X_t$ is larger than $\theta_k + \epsilon^{\mathrm{adv}}$, we have $\mathbb{P}(Y_{t+1} = +1) = p$ and $\mathbb{P}(Y_{t+1} = -1) = 1 - p$. One of important observations is when the query is outside of $\mathbb{B}(\theta_k)$, the gradient information provided by the oracle will have the same probability measure for all functions in $\mathcal{F}'$. This implies

$$D_{\mathrm{KL}}\big(\mathbb{P}(Y_t \mid M, X_t \notin \mathbb{B}(\theta_k), \xi_k) \| \mathbb{P}(Y_t \mid M', X_t \notin \mathbb{B}(\theta_k), \xi_k)\big) = 0, \quad \forall f_M, f_{M'} \in \mathcal{F}'(\theta_k). \tag{32}$$

On the other hand, when $X_t \in \mathbb{B}(\theta_k)$, for any $f_M, f_{M'} \in \mathcal{F}'(\theta_k)$, we can upper bound the KL divergence as follows:

$$D_{\mathrm{KL}}\big(\mathbb{P}(Y_t \mid M, X_t \in \mathbb{B}(\theta_k), \xi_k) \| \mathbb{P}(Y_t \mid M', X_t \in \mathbb{B}(\theta_k), \xi_k)\big) = p \log \frac{p}{1-p} + (1-p) \log \frac{1-p}{p}.$$

Thus, according to Lemma 3, we have:

$$I(M; \widehat{M}_T | \xi_k) \leq c(p) \sum \mathbb{P}(X_t \in \mathbb{B}(\theta_k) | \xi_k), \tag{33}$$

where $c(p) = (2p - 1) \log(p/(1 - p))$. Putting together the pieces yields our result. $\qquad\square$

## B.3 Proofs for Theorem 4

*Proof.* We first prove the result for point error, the result of function error can be achieved by a Jensen's inequality (please see the end of the proof). The general technique of our proof is rather similar to that of statistical minimax analysis for oracle complexity in stochastic convex optimization, but the construction here is a bit more intricate. Specifically, we will pick two similar functions $\mathcal{F}' = \{f_1, f_2\}$ in the class $\mathcal{F}$ and show that they are hard to differentiate with only $T$ queries to the oracle $\phi^{(1)}$. A significant difference to the standard minimax function construction, as will be shown shortly, is that the way how we construct such $f_1$ and $f_2$: our goal is to make the information gain on differentiating $f_1$ and $f_2$ will be zero as long as the learner queries the points a bit far from optimal points. In particular, consider the domain $\mathcal{X} = [0, 1]^d$, we first define following base functions, which will be used for us to construct $f_1$ and $f_2$: $f_0(x) = c_0 \|x - x_0^*\|^\kappa$; $h_1(x) = c_1 \|x - (x_0^* - \epsilon/\sqrt{d} \cdot \mathcal{I}_d)\|^\kappa + c_2$, and $h_2(x) = c_1 \|x - (x_0^* + \epsilon/\sqrt{d} \cdot \mathcal{I}_d)\|^\kappa + c_2$, where $x_0^* = (1/2, \ldots, 1/2)$. We now define functions $f_1$ and $f_2$ as follows:

$$f_1(x) = \max\{f_0(x), h_1(x)\}; \quad f_2(x) = \max\{f_0(x), h_2(x)\}, \tag{34}$$

where $c_0, c_1$ are constants ensuring $f_1$ and $f_2$ are $L$-Lipschitz. Convexity is maintained by the maximum operator over two convex functions. Let $x'$ be one of the solutions for $f_0(x) = h_2(x)$,

Figure 2: A illustration for construction of Convex functions when $\kappa = 2$ and $d = 1$. (a) We use functions $f_0(x) = 0.5|x - 3|^2$, $h_1(x) = 0.2|x - (3 - 0.5)|^2 - 1.6$ and $h_1(x) = 0.2|x - (3 + 0.5)|^2 - 1.6$ as base functions. (b) We then construct $f_1(x) = \max\{f_0(x), h_1(x)\}$ and $f_2(x) = \max\{f_0(x), h_2(x)\}$. In this plot, we choose these numerical constants to ensure that the functions $f_1$ and $f_2$ are indistinguishable based only the function and gradient information when the query points are outside adversary's estimation region.

which $x'$ should depend on the constant $c_2$. We now chose $c_2$ to satisfy following condition: $\|x_0^* - x'\| \geq \epsilon^{\mathrm{adv}}$. By and large, $f_1$ are $f_2$ are constructed such that the learner has to *strenuously* nail down her search within a region which is near to the minimizer. Note that, even though we only construct two functions in $\mathcal{F}'$, we can still ensure that each estimation ball $\mathbb{B}(\theta_k)$ (e.g., when $d = 1$, the subinterval $[2(k-1)\epsilon^{\mathrm{adv}}, 2k\epsilon^{\mathrm{adv}}]$), for adversary, contain the same number of hypothesis functions we construct. To see this, we can just add one more randomization before the Nature draws function $f \in \mathcal{F}$. In particular, we can just replicate a same function subclass for each estimation ball by translating the above $\mathcal{F}'$ along the domain $\mathcal{X}$. Thus, the Nature can just first uniformly sample a function subclass, then sample a function $f$ from that subclass. By construction, we can ensure the quantity $\mathcal{F}'(\theta_k) = 2$ for each estimation ball $\mathbb{B}(\theta_k)$.

Also, note that by triangle inequality, upon defining $\pi(f_1, f_2) = \|x_{f_1}^* - x_{f_2}^*\|$ will guarantee us the property in (3). Moreover, let $J$ denote the region $J = \{x : x \in \mathbb{B}(x_0^*, \|x_0^* - x'\|)\}$ which may contain the ball $\mathbb{B}(\theta_k)$ (this is by our condition for $c_2$). Noticeably, the function $f_1(x)$ and $f_2(x)$ are different only within the region $J$, while they are indistinguishable based only on function value and gradient information calculated outside $J$.

We now proceed to utilize the information bounds we derive in earlier sections to prove our main result. Note that, by construction, at most two functions whose $x^*$s will locate in the region $J$, same for $\mathbb{B}(\theta_k)$. Thus, given the realized selected function index $M \in \{1, 2\}$, by Fano's inequality, we have

$$I(M; \widehat{M}_T | \xi_k) \geq \log 2 - h_2(\delta), \tag{35}$$

where $h_2(\delta) := -\delta \log \delta - (1 - \delta) \log(1 - \delta)$ is the binary entropy function. Let $\mathbb{Q} = \frac{1}{2} \sum_{i=1}^{2} \mathbb{P}(Y_t | M = i, X_t)$, we then have

$$I(M; \widehat{M}_T | \xi_1) \leq \sum_{t=1}^{T} \sum_{x \in \mathcal{X}} \mathbb{P}(X_t = x) \cdot \mathbb{E}_{M,M'} D_{\mathrm{KL}} \left( \mathbb{P}(Y_t | M, X_t = x, \xi_k) \| \mathbb{P}(Y_t | M', X_t = x, \xi_k) \right).$$

Note that by Lemma 3, the RHS of the above inequality can be divided into two parts: one is for summation over $x \notin \mathbb{B}(\theta_k)$, while another is for $x \in \mathbb{B}(\theta_k)$.

By construction we know that $f_1$ and $f_2$ are indistinguishable when $x \notin J$, the same holds for $x \in \mathbb{B}(\theta_k)$. Thus, the learner will obtain no information on which function she is optimizing if her queries are outside of the domain $J$. In other words, the KL divergence will equal to zero when $x \notin J$:

$$D_{\mathrm{KL}} \left( \mathbb{P}(Y_t | M, X_t = x, \xi_k) \| \mathbb{P}(Y_t | M', X_t = x, \xi_k) \right) = 0, \quad \forall x \notin J. \tag{36}$$

We now proceed to bound the divergence $D_{\mathrm{KL}}(\mathbb{P}(Y \mid M, X, \xi_k) \| \mathbb{P}(Y \mid M', X, \xi_k))$ when $x \in J$. Recall that the response from the oracle at the query point $x$ contains the value of $f(x)$ and its gradient information at $x$: $g(x)$. In particular, let $y_1 = f(x) + w_1$ and $y_2 = g(x) + w_2$ denote the noisy function value and noisy gradient value, respectively. Then $y_1$ and $y_2$ are conditionally independent given $M = i$ and $X = x$, for the Gaussian oracle, they can be represented as follows:

$$\mathbb{P}(Y \mid M = i, X = x, \xi_k) = \mathbb{P}(y_1 \mid M = i, X = x, \xi_k) \cdot \mathbb{P}(y_2 \mid M = i, X = x, \xi_k),$$

where $\mathbb{P}(y_1 \mid M = i, X = x, \xi_k) = \mathsf{N}(f_i(x), \sigma^2)$ and $\mathbb{P}(y_2 \mid M = i, X = x, \xi_k) = \mathsf{N}(g_i(x), \sigma^2 \mathcal{I}_d)$, and $\mathcal{I}_d$ denotes a $d-$dimensional identity vector. Therefore, we can bound the divergence

$$
\begin{aligned}
&D_{\mathrm{KL}}\big(\mathbb{P}(Y|M = i, X = x, \xi_k) \| \mathbb{P}(Y|M = j, X = x, \xi_k)\big) \\
=&D_{\mathrm{KL}}\big(\mathbb{P}(y_1|M = i, X = x, \xi_k) \| \mathbb{P}(y_1|M = j, X = x, \xi_k)\big) + \\
&D_{\mathrm{KL}}\big(\mathbb{P}(y_2|M = i, X = x, \xi_k) \| \mathbb{P}(y_2|M = j, X = x, \xi_k)\big) \\
=&D_{\mathrm{KL}}\big(\mathsf{N}(f_i(x), \sigma^2) \big\| \mathsf{N}(f_j(x), \sigma^2)\big) + D_{\mathrm{KL}}\big(\mathsf{N}(g_i(x), \sigma^2 \mathcal{I}_d) \big\| \mathsf{N}(g_j(x), \sigma^2 \mathcal{I}_d)\big) \\
=&\frac{1}{2\sigma^2}\left([f_i(x) - f_j(x)]^2 + \|g_i(x) - g_j(x)\|^2\right).
\end{aligned}
$$

Take the supreme over $\mathcal{X}$ and all possible $(i, j)$ conditional on the event $\xi_k$ will yield us following:

$$
\begin{aligned}
&D_{\mathrm{KL}}\big(\mathbb{P}(Y|M = i, X = x, \xi_k) \| \mathbb{P}(Y|M = j, X = x, \xi_k)\big) \\
&\leq \sup_{\substack{x \in \mathcal{X}; \\ f_i, f_j \in \mathcal{F}'(\theta_k)}} \frac{1}{2\sigma^2}\left([f_i(x) - f_j(x)]^2 + \|g_i(x) - g_j(x)\|^2\right).
\end{aligned}
$$

Thus, back to Lemma 3, we have

$$
\begin{aligned}
&I(M; \widehat{M}_T | \xi_k) \\
\leq& \sum_{t=1}^{T} \mathbb{P}(\|X_t - x^*\| \leq \epsilon^{\mathrm{adv}}) \cdot \max_{x \in J} \frac{[f_1(x) - f_2(x)]^2 + \|g_1(x) - g_2(x)\|^2}{\sigma^2} \\
\leq& \frac{c_1^2}{\sigma^2} \max_{x \in J} \left(\left(\left\|x_0^* - \frac{\epsilon \mathcal{I}_d}{\sqrt{d}}\right\|^\kappa - \left\|x_0^* + \frac{\epsilon \mathcal{I}_d}{\sqrt{d}}\right\|^\kappa\right)^2 + \kappa^2 \left(\left\|x_0^* - \frac{\epsilon \mathcal{I}_d}{\sqrt{d}}\right\|^{\kappa-1} - \left\|x_0^* + \frac{\epsilon \mathcal{I}_d}{\sqrt{d}}\right\|^{\kappa-1}\right)^2\right). \\
&\sum_{t=1}^{T} \mathbb{P}(\|X_t - x^*\| \leq \epsilon^{\mathrm{adv}}) \\
\leq& \mathcal{O}\left(\frac{\epsilon^{2\kappa-2}}{\sigma^2}\right) \sum_{t=1}^{T} \mathbb{P}(\|X_t - x^*\| \leq \epsilon^{\mathrm{adv}}).
\end{aligned}
$$

As a consequence, we have following:

$$
\sum_{t=1}^{T} \mathbb{P}(\|X_t - x^*\| \leq \epsilon^{\mathrm{adv}}) \geq \mathcal{O}\left(\frac{\sigma^2(\log 2 - h_2(\delta))}{\epsilon^{2\kappa-2}}\right) \tag{37}
$$

Putting together our bounds with the Equation (9) will give us desired secure oracle complexity for point error. For the result of function error, note that given $\kappa > 1$ we have

$$
\inf_{\mathcal{A}} \sup_{f \in \mathcal{F}'} \mathbb{E}\left[|f(\widehat{X}_T) - f^*|\right] \geq \inf_{\mathcal{A}} \sup_{f \in \mathcal{F}'} \mathbb{E}[\lambda \|\widehat{X}_T - x_f^*\|^\kappa]. \tag{38}
$$

$$
\geq \inf_{\mathcal{A}} \sup_{f \in \mathcal{F}'} \mathbb{E}[\lambda \|\widehat{X}_T - x_f^*\|]^\kappa. \qquad \text{(By Jensen's inequality)}
$$

Invoking Markov inequality will give us the secure oracle complexity for function error. $\qquad \square$

Figure 3: A graph illustration on Algorithm 1. The length of each shade interval is $2\epsilon$. The black dots $\{X_1, X_2, \ldots\}$ are the queries for last phase, while the red dot is the location of minimizer.

---

**Algorithm 1** Secure Learning Protocol

1: **Input:** $S := \lfloor 1/\delta^{\mathrm{adv}} \rfloor$, $K := \lfloor T/S \rfloor$, exponent $\kappa > 0$, convexity parameter $\lambda > 0$, confidence $\delta > 0$, subgradient bound $W$.
2: Initialize $x_1 \in [0,1]$ arbitrarily and set $\mathcal{G}_1 = \{g_1\}$, divide $[0,1]$ into subintervals with the equal length of being $\delta^{\mathrm{adv}}$.
3: **for** $k = 1, \ldots, K$ **do**
4:     Let $\bar{x} = \mathrm{EpochGD}(\kappa, \lambda, \delta, W, K, \mathcal{G}_k)$.
5:     **for** $s \in [S]$ **do**
6:         Query the oracle at the point $x_{(k-1)S+s+1} = (s-1)\delta^{\mathrm{adv}} + \left(\bar{x} - \left(J(\bar{x}, \delta^{\mathrm{adv}}) - 1\right)\delta^{\mathrm{adv}}\right)$.
7:         **if** $s = J(\bar{x}, \delta^{\mathrm{adv}})$ **then**
8:             Record the gradient $g_{(k-1)S+s+1}$ obtained from the oracle: $\mathcal{G}_k \leftarrow \mathcal{G}_k \cup \{g_{(k-1)S+s+1}\}$.
9:         **end if**
10:     **end for**
11: **end for**
12: **Output:** Learner's estimation: $\bar{x}$.

---

### B.4 Algorithm and the Proof for Theorem 5

For notational simplicity, let $J(x, \delta^{\mathrm{adv}})$ denote the index of subinterval which contains the point $x$ when $[0,1]$ is uniformly divided in subintervals with the length of $\lfloor 1/\delta^{\mathrm{adv}} \rfloor$ and let $K = \lfloor T/S \rfloor$.

*Proof.* We now establish the privacy guarantee when the adversary's error measure is point error. The proof can be similarly carried over to function error. Recall that the learner actually performs parallel EpochGD on the $S$ subintervals $\{((s-1)\delta^{\mathrm{adv}}, s\delta^{\mathrm{adv}}]\}_{s \in [S]}$. Since the adversary only observes the queries, and he is not aware of the learner's confidential computation oracle, he learns that $X^*$ is contained in one of these $S$ subintervals. Moreover, due to the strictly symmetrical querying over these subintervals, the adversary also cannot tell which of the subintervals contains $X^*$. Specifically, let $\{X_s\}_{s \in [S]}$ denote the learner's last phase queries. Then the adversary knows following:

$$\frac{1-\delta}{S} \leq \mathbb{P}(|X_s - X^*| \leq \epsilon) \leq 1, \quad \forall s \in [S].$$

Thus, at the end of the last phase, the adversary will know that $X^*$ belongs to one of the subintervals $\{[X_s - \epsilon, X_s + \epsilon]\}_{s \in [S]}$ with high probability, where $\epsilon = \tilde{\mathcal{O}}\left((T\delta^{\mathrm{adv}})^{-\frac{1}{2\kappa-2}}\right)$. Without loss of generality, assume the adversary is endowed with an uniform prior knowledge on where $X^*$ is and assume the maximum uncertainty for $X^*$. Then it can be computed that the adversary's posterior density of $X^*$ is:

$$f_{X^*}(x|\text{queries}) = \begin{cases} (1-\delta)/(2S\epsilon), & \forall x \in \cup_{s=1}^{S}[X_s - \epsilon, X_s + \epsilon] \\ \delta/(1 - 2S\epsilon), & \text{o.w.} \end{cases} \tag{39}$$

Since $\delta$ is a small value (i.e., $\ll 0.5$), thus, for any subinterval $\mathcal{L} \subset [0,1]$ with the length of $2\epsilon^{\mathrm{adv}}$, it is adversary's best strategy to narrow down his estimation region which could cover one of subintervals $\{[X_s - \epsilon, X_s + \epsilon]\}_{s \in [S]}$. Now, let $\mu(\cdot)$ denote the Lebesgue measure of subsets of $[0,1]$, note that

$$\mu\left(\mathcal{L} \cap \cup_{s=1}^{S}[X_s - \epsilon, X_s + \epsilon]\right) \leq 2\epsilon.$$

Together with the Eqn. (39), we find that, for any adversary's estimator $\widehat{X}^{\mathrm{adv}}$, we have

$$\mathbb{P}(|\widehat{X}^{\mathrm{adv}} - X^*| \leq \epsilon^{\mathrm{adv}}|\text{queries}) \leq \frac{1-\delta}{2S\epsilon} \cdot 2\epsilon + \frac{\delta}{1 - 2S\epsilon} \cdot (2\epsilon^{\mathrm{adv}} - 2\epsilon). \tag{40}$$

---

**Algorithm 2** EpochGD $(\kappa, \lambda, \delta, W, T, \mathcal{G})$

---

1: Initialize $x_1^1 = x_1$, $e = t = 1$.
2: Initialize $T_1 = 2C_0$, $\eta_1 = C_1 \, 2^{-\frac{\kappa}{2\kappa-2}}$, $R_1 = \left(\frac{C_2 \eta_1}{\lambda}\right)^{1/\kappa}$.
3: **while** $\sum_{i=1}^{e} T_i \leq T$ **do**
4:     **if** $|\mathcal{G}| < \sum_{i=1}^{e} T_i$ **then**
5:         Get the newest element in $\mathcal{G}$, denote it by $g_t$.
6:         Set $\mathcal{K} := [0,1] \cap [x_1^e - R_e, x_1^e + R_e]$.
7:         **Output:** $x_{t+1}^e = \arg\min_{x \in \mathcal{K}} |(x_t^e - \eta_e g_t) - x|$.         ▷ Return the value to protocol
8:         **Update:** $\mathcal{G}$.
9:         Set $t \leftarrow t + 1$.
10:     **else**
11:         Set $x_1^{e+1} = \frac{1}{T_e} \sum_{t=1}^{T_e} x_t^e$.
12:         **Output:** $x_1^{e+1}$.         ▷ Return the value to protocol
13:         **Update:** $\mathcal{G}$.
14:         Set $T_{e+1} = 2T_e$, $\eta_{e+1} = \eta_e \cdot 2^{-\frac{\kappa}{2\kappa-2}}$.
15:         Set $R_{e+1} = \left(\frac{C_2 \eta_{e+1}}{\lambda}\right)^{1/\kappa}$, $e \leftarrow e + 1$, $t = 1$.
16:     **end if**
17: **end while**
18: **Output:** $x_1^e$.

---

Under the assumption that $2\epsilon^{\text{adv}} < \delta^{\text{adv}}$, the RHS of Eqn. (40) will be smaller than $1/S$. We thus establish the privacy guarantee for any adversary's estimators.

We prove the accuracy guarantee of the above secure learning protocol with appropriate chosen $C_0, C_1, C_2$. Specifically, set $C_0 = 288 \log(\lfloor \log T \delta^{\text{adv}} + 1 \rfloor / \delta)$, $C_1 = \frac{G^{\frac{2-\kappa}{\kappa-1}} 2^{\frac{\kappa}{2(\kappa-1)^2}}}{\lambda^{1/(\kappa-1)}}$, $C_2 = 2^{\frac{\kappa}{2\kappa-2}} W^2$. Follow the analysis of [7, 12], we know that given a total oracle budge $T$ with dividing into a series of consecutive epochs $\{T_1, 2T_1, \ldots, 2^e T_1, \ldots\}$, and running standard stochastic gradient descent in each epoch, will ensure us $f(\widehat{X}_T) - f^* \leq \tilde{\mathcal{O}}\left(T^{-\frac{\kappa}{2\kappa-2}}\right)$ and $|\widehat{X}_T - X^*| \leq \tilde{\mathcal{O}}\left(T^{-\frac{1}{2\kappa-2}}\right)$ hold with probability at least $1 - \delta$ for some estimator $\widehat{X}_T$. Thus, adapted to our setting, our total oracle budget is $\lfloor T \delta^{\text{adv}} \rfloor$. Plugging this into the above results will help us to get the accuracy guarantee. As a sanity check, one can also verify that the error rate presented in our Theorem 5 can be easily translated to match our oracle complexity in Theorem 4. $\qquad\square$