[Reviews · NeurIPS 2020]

Review 1

Summary and Contributions: This paper investigates the secure stochastic convex optimization problem. In this model, a learner seeks to minimize a convex function by interacting with a stochastic first-order oracle, and there is an adversary who only observes the queries made by the learner. The goal of both learner and adversary are to minimize the objective, but the learner also wants to prevent the adversary from achieving high accuracy. For this problem, a generic information-theoretic method is provided to prove lower bounds for secure stochastic convex optimization. This method is based on a particular adversary, called the proportional sampling estimator (proposed in previous work), which simply chooses uniformly at random among all iterates of the learner. This method is applied to both binary search (deterministic and stochastic) and stochastic convex optimization in 1 and higher dimensions. A matching upper bound for one-dimensional case is also provided (which is not sufficiently discussed in the main file). %%%%%%%% AFTER REBUTTAL %%%%%%%%%%%%%%%% I have reviewed the authors response, and I am maintaining my overall score.

Strengths: This is a well-posed and interesting theoretical problem; in terms of their practical consequences, they are not entirely clear, but I don't see this as a problem (this is a problem established in the past couple of years, and those papers have set the motivations behind this problem clear). The information theoretic methods proposed, despite leveraging substantially from past work, provide innovative ways to incorporate "security" for the learner. I believe these results will be well received and appreciated by the NeurIPS community.

Weaknesses: On the downside, the paper is not very clear in some details, particularly regarding sharpness of the bounds and dimension dependence.

Correctness: The methods are correct, as far as I could verify (I verified parts of the supplementary file and all the main document). However, I belive the conclusions suffer from some gaps, particularly regarding sharpness of the upper and lower bounds, particularly in high-dimensional settings (more comments below).

Clarity: Aside from the aforementioned points, and those made below, the paper is well written.

Relation to Prior Work: The comparison from past work seems fairly thorough.

Reproducibility: No

Additional Feedback: High-level Comments: 1. In terms of the presentation, the paper constantly switches in terminology, both referring to "secure" and "private." Given that by now differential privacy has been established as a notion of formal privacy, I believe it is better to refer to this problem to "secure" optimization. For example, in page 2, line 49, classic bounds are referred as "non-private." Similarly to Definition 2 ((eps,delta)-private), Definition 3 (private query complexity), etc. I think it would be better to exclusively refer to "secure" in this definitions. 2. The lower bounds, particularly those for binary search, are claimed to be tight. However, they are clearly vacuous for the classical "non-secure" case. For this, choose \eps^{adv}=2\eps and \delta^{adv}=1; this results in a constant lower bound (although binary search requires \log(1/\eps) queries). Other works, such as (Xu & Yang:2019) can at least reproduce the non-secure bound (by using in the numerator of the \log\max\{\eps^{adv},2^{-L}\}. Can the authors comment on this? 3. Page 4, line 136. The Lipschitzness and uniform convexity properties are restricted to $\|\cdot\|_2$? If so, please make it explicit. This comment is not minor, since the stochastic convex optimization problem w.r.t. $\ell_{\infty}$ ball domain and $\ell_2$-Lipschitz constant does have a dimension dependent lower bound (see Thm. 1 in Agarwal et al. 2012 paper). This is very far from the lower bound proved in Thm. 4. 4. The proof of Thm. 5 is only obtained for the one-dimensional case. This is not explicit in the statement of the Theorem (even though it is stated in the preceeding paragraph). 5. Would it be possible to briefly explain the epoch GD algorithm, either in the main text or supplementary files? The discussion here seems hasty. 6. Regarding information-theoretic methods for the complexity of optimization, perhaps the authors could be interested in https://arxiv.org/pdf/1407.5144.pdf. Techniques from that paper are very likely useful to establish query complexity for secure non-stochastic convex optimization. Detailed comments: 1. Page 5, line 182. "leaner" should be "learner." 2. Page 7, Thm 4. "Secure Sotchastic Convex Optimization" should be "Stochastic." 3. Page 7, line 287. "The convergence for point error would fail with non-strongly convex functions." What does this mean? Are you referring to the fact that the lower bound grows to infinity? Please re-phrase. 4. Supplementary material, proof of Lemma 1. I don't understand why after finishing the proof (in line 402) the proof continues. 5. Supplementary material, page 2, line 429, I don't understand why M and Y_t are conditionally independent: Is this because $\xi_k$ determines the oracle information? Please give a more detailed explanation here.


Review 2

Summary and Contributions: The authors study the secure stochastic convex optimization problem: a learner aims to learn the optimal point of a convex function through sequentially querying a (stochastic) gradient oracle, in the meantime, there exists an adversary who aims to free-ride and infer the learning outcome of the learner from observing the learner’s queries. The adversary observes only the points of the queries but not the feedback from the oracle. The goal of the learner is to optimize the accuracy, i.e., obtaining an accurate estimate of the optimal point, while securing her privacy, i.e., making it difficult for the adversary to infer the optimal point. The authors quantify this tradeoff between learner’s accuracy and privacy and characterize the lower and upper bounds on the learner’s query complexity - i.e., minimum number of samples needed from a learner, as a function of desired levels of accuracy and privacy. They provide a general template based on information theoretical analysis and show lower bounds for stochastic convex optimization and (noisy) binary search. They also present a generic secure learning protocol that randomly samples points in well-spaced intervals, and achieves the matching upper bound up to logarithmic factors.

Strengths: The problem setting is very relevant to machine learning, and a less-studied one. The key ideas introduced in the paper are intuitive -- sample a random set of necessary and unnecessary points to confuse the adversary about multiple functions with well-spread out minimizers. ------------------------------------------ EDIT: I thank the authors for the response to reviews, but I am inclined to maintain my score.

Weaknesses: 1. Applications: In the dynamic pricing application stated in the paper, once a learner estimates the demand function and optimal price for maximum revenue, the adversary will learn this price since learner will use the optimal price eventually. Can the learner not use the optimal price? Moreover, it is illegal to chase prices based off competitor prices. Are there other applications where this problem setting would be useful? 2. Problem definition: Why are $\epsilon$ and $\epsilon_adv$ two different parameters, similar why are $\delta$ and $\delta_adv$ different? It might be good to discuss an impossibility example where they cannot be same (if that is the case). Or discuss relation between the two - $\epsilon_adv$ should be lower than $\epsilon$. This discussion comes up a little bit towards the end of the paper, but making this more transparent would be better. 3. Constructing difficult instances: The authors assume a lot about the scale of the function and spread of minimizers and therefore the problem -- the minimizers are one of an epsilon grid. Can this parametric class of functions with \epsilon distance be chosen without assuming anything about the spread of the function? How dependent or independent is this to the assumptions on the set of potential minimizers? 4. Weakness of the adversary: The adversary uses only a proportional sampling estimator -- that seems a bit too weak -- as it does not consider the trajectory of the points played by the learner. Also, what is unclear to me is if the learner needs to continue sampling points to throw off the adversary? There should be either a time limit on the process, or the analysis should hold for "any time t" wherein the learner desires to continue to have at least some fraction of optimum objective on average, and the adversary continues to only proportionally sample the last few sequence of points. 5. Sampling points: Sampling an $\epsilon$ grid of the solution space (in a pre-determined manner) will not reveal any information to the adversary. This would be a completely deterministic strategy. Where is randomization useful?

Correctness: I think so, but I have only skimmed through the appendix.

Clarity: The paper is well written, mostly. Here are some points the authors should pay attention to: 3. Many grammatical errors throughout the paper -- ``with correct probability" lines 122, Line 190 - "the adversary first identify". Typos - Line 182 ``leaners optimization problem" instead of learners. What does this mean ``we adopt them to prove the complexity" in line 195. 4. Raising mathematical quantities to numeric superscripts is awkward, e.g. $\phi^1$ or $\phi^2$. Notation is quite bad: $f^\prime$ is used, and this does not mean derivative, even though the authors talk about one-dimensional functions. 5. Line 149 -- ``accurate optimization algorithms" -- what does this mean? You cannot, typically, obtain "accurate" convex optimization methods - there are almost always approximate -- even the ones that converge linearly! 6. Line 187 - "While incorporating a weaker adversary could lead to weaker lower bounds, as we demonstrate later" -- do the authors mean stronger adversaries (who go beyond proportional sampling estimators?)? 7. Lines 193 - what is \theta_1, \theta_2, ..\theta_k -- I suppose the authors mean equally distance points by \epsilon_adv/L distance but this is not defined anywhere! 8. Equation (6) the function I(\cdot) is never defined but used in the theorem. I suspect it means conditional mutual information? 9. In Lemma 1, should T be T_P?

Relation to Prior Work: Yes, except "how different" their analysis is from a previous Private Binary Search paper is unclear.

Reproducibility: Yes

Additional Feedback:


Review 3

Summary and Contributions: This paper studies the secure stochastic convex optimization problem where a learner aims to learn the optimal point of a convex function through sequentially querying a (stochastic) gradient oracle. In the meantime, there exists an adversary who aims to free-ride and infer the learning outcome of the learner from observing the learner's queries but not the feedback from the oracle. The goal of the learner is to obtain an accurate estimate of the optimal point, while securing her privacy, i.e., making it difficult for the adversary to infer the optimal point. The tradeoff between learner’s accuracy and privacy is characterized by the lower and upper bounds on the learner's query complexity as a function of desired levels of accuracy and privacy. This paper also presents a generic secure learning protocol that achieves the matching upper bound up to logarithmic factors.

Strengths: This paper characterizes the fundamental trade-offs between learner’s accuracy and privacy using query complexity (the minimum number of queries needed to achieve a given level of accuracy and privacy) in the PAC framework, and provides a generic secure learning protocol that achieves the matching upper bound up to logarithmic factors. I believe the problem studied in the paper can be applied in various application, including distributed learning or federated learning system, which would attract sufficient audience in NeurIPS. EDIT: I thank the authors for the response to reviews, and I am maintaining my overall score.

Weaknesses: 1. I think one major issue about the paper is the organization. In the main body of the paper, a lot of efforts are spent to emphasize the proof of lower bound. However, it is said in the paper that the proof technique leads to a weaker lower bound (reduce from the set F to F’ by Le Cam’s method), it can be shown that the lower bound is tight only when a matching upper bound can be obtained. However, the description in Section 4 is not enough for readers to understand the details about the upper bound and the optimal secure learning protocol. Some adjustments need to be done.

Correctness: The proofs looks reasonable to me.

Clarity: The organization of the paper needs to be improved.

Relation to Prior Work: Prior works are addressed well.

Reproducibility: Yes

Additional Feedback:


Review 4

Summary and Contributions: This paper is about stochastic optimization with guarantees on a new privacy measure. The new privacy measure is defined as follows. There is an adversary who uniformly samples a point from \{ x_1, x_2, ... x_T \}, where x_t's are the iterates maintained by an underlying optimization algorithms throughout T iterations (line 189). Denote \hat{x}_{adv} the sampled point. A stochastic optimization algorithm with a good privacy measure is defined in the sense that the function value at \hat{x}_{adv} is not too close to the optimal value (line 106).

Strengths: Some theoretical analysis is available for the problem, which is by leveraging the techniques developed by [Agarwal et al. 2009]. Specifically, the paper provides a lower bound result (e.g. how many iterations does an algorithm need in order to achieve \epsilon-private) as well as an upper bound result.

Weaknesses: Here are my concerns about this work: (1) A fundamental question is about the implications of this new privacy measure. What is the point of defining such a measure? The authors propose the new privacy measure, which is very different from the related works of differential private optimization (e.g. [Feldman et al 2020] and [Bassily et al. 2019]). The new measure seems not a standard in the literature and I don't see any discussions on its implications neither do I see a comparison with other existing privacy measures. Ref: [1] Private Stochastic Convex Optimization: Optimal Rates in Linear Time. Vitaly Feldman, Tomer Koren, and Kunal Talwar. STOC 2020 . [2] Private Stochastic Convex Optimization with Optimal Rates. R. Bassily, V. Feldman, K. Talwar, A. Thakurta, NeurIPS 2019 (2) It is well known in the stochastic optimization literature that the convergence is achieved either by *averaging* or by doing something to ensure the *last iterate* enjoy some guarantees under certain conditions like strong convexity (see ref. [3] [4] [5] [6]). In other words, the function value at any iterate during stochastic optimization cannot be too close to the optimal value due to the noise. So the results here seems to be expected, based on the *privacy* measure defined in this paper. Ref: [3] Boris T. Polyak, Anatoli B. Juditsky. Acceleration of Stochastic Approximation by Averaging. SIAM Journal on Control and Optimization. 30(4):838-855, 1992. [4] Francis Bach and Eric Moulines. Non-strongly-convex smooth stochastic approximation with convergence rate. NeurIPS 2013. [5] Making the Last Iterate of SGD Information Theoretically Optimal Prateek Jain, Dheeraj Nagaraj, Praneeth Netrapalli. COLT 2019. [6] Arkadii Nemirovskii and David Borisovich Yudin. Problem Complexity and Method Efficiency in Optimization. Wiley, 1983.

Correctness: seems correct

Clarity: seems ok

Relation to Prior Work: see *weakness*

Reproducibility: Yes

Additional Feedback: *** after rebuttal *** Thanks to the response. I upgrade my score.

[Author Response · NeurIPS 2020]

We thank all the reviewers for the thoughtful comments and suggestions. We'll fix typos and make descriptions clear.

**@R1, sharpness of the bounds and dimension dependence:** Our work focuses on understanding the learner's privacy-
efficiency trade-off. Our current results give the query complexity bounds that depend on the desired level of privacy
and accuracy. Obtaining a dimension-dependent bound is an important, and ongoing research.
*Lower bounds for secure binary search:* We'd like to first clarify that our setting is more towards a Bayesian setting.
The bound cited by the reviewer is for a deterministic setting. To compare with [17] (we use the citation number in
our submission in this response, [17] is Xu & Yang:2019) in the Bayesian setting, in our proof, we use a Proportional-
Sampling (PS) estimator as in [18] (Xu 2018) to characterize the hardness results, while the authors in [17] use a
different adversary (*truncated* PS estimator) and obtain sharper bounds. While ours are tight when the learner's error
$\epsilon \to 0$ given a fixed $(\epsilon^{\mathrm{adv}}, \delta^{\mathrm{adv}})$. For a concise presentation, our results also hide logarithmic factors which could make
a difference in these cases ($\epsilon \to 0$). We will add more discussions and make statements more rigorous.

**@R1, terminology, notations and detailed comments:** *"secure" vs "private":* We agree that "secure" would be a
better choice to deliver our main message and will update accordingly. $\ell_2$-*norm:* Yes, our Lipschitzness and uniform
convexity properties are defined w.r.t $\ell_2$-norm. We recognize a dimension-dependent bound as an imporant future
direction. *"The convergence for point error would fail ...":* Yes, we meant to say that the lower bound cannot be
bounded. We'll rephrase it. *Line 402, the proof of Lemma 1:* The proof of Lemma 1 should finish at Line 402, as
Lemma 1 is stated about function error. The proof then continues (Line 402 – 408) for point error. *Line 429, conditional
independency of $M$ and $Y_t$:* $Y_t$ and $M$ are conditionally independent given the information $(X^t, Y^{t-1}, \xi_k)$, in other
words, $M \to (X^t, Y^{t-1}, \xi_k) \to Y_t$ is a Markov chain. We will make these statements/terminology clear.

**@R2, applications:** We focus on settings in which the learner cares about the accuracy of the their final estimate instead
of the utility generated during learning. For example, in federated learning (the example used in [17]), companies may
optimize the parameters of their learning models using gradient decent through sequentially broadcasting their models
to data-holding users and obtain the gradient information. Adversaries can pretend to be users and estimate the final
model through observing the broadcasted models (but not the gradient). As another example, companies may perform
market research by sequentially inviting users in different population for interview. Competitors might free-ride the
outcome by observing who attended the interview but not the responses.

**@R2, $(\epsilon, \delta)$ and $(\epsilon^{\mathrm{adv}}, \delta^{\mathrm{adv}})$:** These two set of parameters cannot be the same and need to satisfy certain conditions.
(as mentioned in Footnote 5). Intuitively, the learner has more information than adversary, therefore it would make
more sense for adversary to have a weaker requirement. We will include the discussion in the revision.

**@R2, difficult instances and adversary:** To prove the lower bound, one only needs to find a hard problem instance
(and a specific adversary) to prove that *any* algorithm has to use at least sufficient number of queries. While our current
constructions of problem instance and adversary are specific, they serve the purpose of proving a lower bound. While
one might wonder whether we can find constructions leading to tighter lower bound, since we also provide a matching
upper bound, this means that this particular choice of constructions leads to a tight lower bound.

**@R2, other comments:** Randomization helps to ensure the adversary can't do better by guessing uniformly at random.
Pre-defining a deterministic sampling would work as well, but we might need to account for the adversary's prior and
belief. Line 195 – "...we adopt them ...", we mean that the two ways in defining PS estimator are essentially the same.
Line 149 – "...accurate optimization algorithms.", we mean that an algorithm could efficiently optimize the function up
to a small error. Line 187 – "While incorporating a weaker adversary ...", yes, we actually mean stronger adversary
(sorry for the typo). *Notations*: We will carefully revise the notations. $\{\theta_1, \ldots, \theta_k\}$ is defined in Line 190: it represents
a $2r$-packing set with radius $r$. $I(\cdot|)$ denotes the conditional mutual information and $T$ in Lemma 1 should be $T_{\mathcal{P}}$.

**@R4, organization and Sec 4:** Thanks for suggestion! We will adjust the content in Sec 4 to make it more readable.

**@R5, comparisons with Differential Privacy (DP):** Thanks for the references. Our privacy notion builds on recent
works [18, 15, 17] and as [18] points out, these two privacy notions are incompatible. At a high level, DP stands from a
universal perspective, where the output distribution of a mechanism is supposed to be insensitive w.r.t any perturbation
in the input, and thus the privacy of individuals is protected. In contrast, the current privacy measure is more specific and
a goal-oriented one, which captures an adversary's (in)ability to perform a specific statistical inference task: Adversary's
goal is to infer an optimizer from observing learner's queries.

**@R5, results "to be expected":** We respectfully disagree. Under our private learning setting, the adversary also has
the complete knowledge of how learner's algorithm operates. Thus, even though the function value at any iterate during
learning process is not close to the optimal value, as long as the learner uses these function values to derive a solution
under the algorithm and when this is fully informed to the adversary, the adversary can formulate his own estimate for
the optimizer via reverse-engineering the algorithm and the trace of queries. The key to prevent this privacy breach is
that the learner should *obfuscate* her learning strategy.

[Meta-Review · NeurIPS 2020]

This paper was carefully reviewed and discussed by our reviewer panel. The consensus was that this is nice work, the rebuttal had some sway, and the paper can be published in NeurIPS this year. But please do take into account the detailed comments of the reviewers when putting together your camera-ready version.